

# Estimates of sub-national methane emissions from inversion modelling

Sarah Connors[1a], Alistair J. Manning[2], Andrew D. Robinson[1], Stuart N. Riddick[1b], Grant L. Forster[3], Anita Ganesan[4], Aoife Grant[5], Stephen Humphrey[3], Simon O'Doherty[5], Dave E. Oram[3], Paul I. Palmer[6], Robert L. Skelton[7], Kieran Stanley[5], Ann Stavert[5c], Dickon Young[5], Neil R. P. Harris[8]

[1]Centre for Atmospheric Science, University of Cambridge, Cambridge, UK
[2]Met Office, Exeter, UK
[3]National Centre for Atmospheric Science (NCAS), School of Environmental Sciences, University of East Anglia, Norwich, UK
[4] School of Geographical Sciences, University of Bristol
[5]School of Chemistry, University of Bristol, Bristol, UK
[6]School of GeoSciences, University of Edinburgh, Edinburgh, UK
[7]Department of Engineering, University of Cambridge, Cambridge, UK
[8]Centre for Environmental and Informatics, Cranfield University, Cranfield, UK
[a]now at Université Paris Saclay, Paris, 91120, France
[b]now at Department of Civil and Environmental Engineering, Princeton University, NJ 08540, USA
[c]now at CSIRO, Oceans and Atmosphere, Aspendale, Australia

*Correspondence to*: Sarah Connors (sarah.connors@universite-paris-saclay.fr) and Neil Harris (Neil.Harris@cranfield.ac.uk)

**Abstract.** Methane is a strong contributor to global climate change, yet our current understanding and quantification of its sources and their variability is incomplete. There is a growing need for comparisons between emission estimates produced using 'bottom-up' inventory approaches and 'top-down' inversion techniques based on atmospheric measurements, especially at higher spatial resolutions. To meet this need, this study presents using an inversion approach based on the Inversion Technique for Emissions Modelling (InTEM) framework and measurements from four sites in East Anglia, United Kingdom. Atmospheric methane concentrations were recorded at 1-2 minute time-steps at each location within the region of interest. These observations, coupled with the UK Met Office's Lagrangian particle dispersion model, NAME (Numerical Atmospheric dispersion Modelling Environment), were used within InTEM$_{2014}$ to produce methane emission estimates for a 1-year period (June 2013 - May 2014) in this eastern region of the UK (~100 x 150 km) at high spatial resolution (up to 4 x 4 km). InTEM$_{2014}$ was able to produce realistic emissions estimates for East Anglia, and highlighted potential areas of difference from the UK National Atmospheric Emissions Inventory (NAEI). As this study was part of the UK Greenhouse gAs Uk and Global Emissions (GAUGE) project, observations were included within a national inversion using all eleven measurement sites across the UK to directly compare emission estimates for the East Anglia Region. Results show similar methane estimates for the East Anglia region. Methane emissions from Norfolk and Suffolk show good agreement with the estimates in NAEI, with differences of ~5%. Larger differences are found for Cambridgeshire where our estimate is 22.5% lower than that of NAEI. The addition of the EA sites within the national inversion system enabled finer spatial resolution and a decrease in the associated uncertainty for that area. Further development of our approach to include a more robust analysis of the methane concentration in the air entering this region and the uncertainty associated with the resulting emissions would strengthen this inverse method. Nonetheless, our results show there is value in high spatial resolution measurement networks and the resulting inversion emission estimates.





## 1. Introduction

Methane is a potent greenhouse gas (GHG) whose atmospheric concentration has quadrupled over the past 20,000 years and now lies well outside the variability observed over the past 800,000 years in the ice core record (Brook and Buizert, 2018). This rise became appreciable around the time of the Industrial Revolution and continues up to the present day.

There is considerable dispute about what is driving the recent rise with possible causes including tropical wetland expansion, increased fossil fuel emissions, and a decrease in the atmospheric removal rate (e.g., Nisbet et al., 2016; Rice et al., 2016; Rigby et al., 2017; Turner et al., 2017). Anthropogenic emissions (principally fossil fuels, agriculture and waste, and biomass burning) constitute approximately 60% of the current emissions (Saunois et al., 2016) and so reductions in methane emissions are feasible.

Methane has a global warming potential of 28 over a 100 year timescale (Harris et al., 2014) and will play a vital role in any attempt to limit global temperature increase to 1.5°C or even 2.0°C (e.g., Comyn-Platt et al. 2018). There is thus great interest in reducing its emissions and atmospheric concentrations in the near future. An essential part of this is, first, knowing what and where the emissions are (source, location, and magnitude) and, second, knowing that these emissions are reducing. Atmospheric observations are an ideal way of providing evidence for both as, through their addition into

inversion methods, they can identify and quantify emission sources, which then can be monitored over time.

National emission inventories are produced as part of the UNFCCC process, which require nations to submit annual estimates of their GHG emissions. These contain detailed information, often available at sub-national scales, and is produced by so-called 'bottom-up' methods, which provide national inventories for multiple emission source sectors. The calculations involve using defined emission factors based on recommended values or field measurements together with

activity data. The UK's National Atmospheric Emissions Inventory (NAEI - Brown et al. 2018) for methane contains annually averaged estimates on a 1x1 km or 5x5 km grid resolution. Emissions are categorised into different SNAP (Selected Nomenclature for sources of Air Pollution) sectors which include 'agric' (SNAP 10 - agriculture, forestry and land-use change), 'waste' (SNAP 09 - waste treatment and disposal), and 'offshore' (SNAP 05 - extraction and distribution of fossil fuels). While the total uncertainty for UK methane emissions in the NAEI is estimated at 40%., the

sub-national scale the uncertainty is much larger. The NAEI also does not include seasonal variations or natural emissions. Inversion, or 'top-down', techniques provide an alternative way of estimating GHG emissions. Emission fluxes are estimated using atmospheric measurements and a meteorological dispersion model that can simulate source to receptor dispersion. Methane emissions have been estimated using many inversion methods at global (e.g., (Bousquet et al., 2011; Houweling et al., 2014), European (e.g., Bergamaschi et al., 2005, 2018), and national (e.g., Ganesan et al., 2014; Rigby

et al., 2011) scales. These approaches provide an independent way of checking the national inventory totals and can assess emission changes over varying timescales (Brown et al., 2018).

This study, performed as part of the Natural Environment Research Council's Greenhouse gAs UK and Global Emissions project (NERC GAUGE) project (Palmer et al., 2018), explores the possibility of producing top-down methane emission estimates on the sub-national scale to be directly compared with the 2012 NAEI. The approach taken is to make

observations at four sites in Eastern England which are tens of km apart. To achieve this, three additional sites measuring atmospheric methane were installed around an existing site, Tacolneston, which is part of the UK DECC (Deriving Emissions linked to Climate Change) network. Emission estimates were synthesised using a top-down sub-national inversion method developed by the UK Met Office (Manning et al., 2003, 2011). This method is a previous version of the approach known as InTEM (Inversion Technique for Emissions Modelling) as used in Arnold et al. (2018) and will

henceforth be referred to as InTEM$_{2014}$. InTEM$_{2014}$ was chosen for two reasons. First, it has been used to produce annual national methane emissions estimates dating back to 1990 (Manning et al., 2011) and so it provides some traceability to the national estimates. Second, we had experience in adapting it in the development of a novel method to estimate CHBr$_3$



emissions around Malaysia (Ashfold et al., 2014). The initial aim of this project was to establish a 'proof of concept' that InTEM$_{2014}$ (and by implication other inversion schemes) could be used at the sub-national scale.

This paper presents the preliminary findings of the work and discusses ways to improve the current setup. Section 2 describes the methodology underlying the measurements and the inverse modelling used. The results are presented and

discussed in Section 3, with particular emphasis on the causes of uncertainty in the adopted approach. In addition, the results of a model calculation performed as part of an inversion incorporating all of the UK measurements collected within GAUGE and DECC, i.e., with the East Anglian measurements nested within the DECC/GAUGE tall tower network, are presented as a possible way forward.

## 2. Methodology

Our approach requires two main elements: (i) calibrated measurements from the four sites; and (ii) an inversion model to provide estimates of the emissions and their uncertainties.

### 2.1 Measurements

### 2.1.1 Sites

The measurement sites for this trial project were located in East Anglia, United Kingdom. This region was chosen for

three reasons:

a) The relatively flat topography. Turbulence in the boundary layer and low troposphere is hard to model at the high resolution required for this study. East Anglia is flat and low-lying, with a highest elevation of 146 m. Uncertainties in the small scale meteorological turbulence that is parameterised, not explicitly modelled, in the dispersion model (see Section 2.2.1) are reduced in this simpler topography compared to more heterogeneous

terrains. The calculated trajectories are thus in principle more accurate than those calculated for areas of the UK with more complex topography.

b) The existence of gradients in the NAEI emissions fields across East Anglia, provides a better test of the inversion system than would a region with homogenous emissions.

c) Its close proximity to Cambridge, and thus has ease of access to the measurement sites for logistical maintenance

and calibration.

The sites' locations were nested within the pre-existing UK DECC tall tower network, which has since been expanded to include two new tall tower sites established under GAUGE (Stavert et al., 2018). The aim was to develop a stand-alone inversion scheme in the first instance and then to integrate these East Anglian measurements into a UK-wide inversion analysis, so the ability to link the calibration of the East Anglian and national networks was vitally important.

The four measurement locations are shown in Figure 1 and some characteristics of the sites are given in Table 1. Sites 1 (Haddenham) and 4 (Tilney) are churches in villages away from the national gas grid and have inlets ~ 25 m above the ground to reduce the influences of local methane sources. The Weybourne Atmospheric Observatory (Site 3, hereafter Weybourne) is coastal, to the North of East Anglia, and has a 10 m mast for its inlet. Two instruments were run in tandem at the Weybourne site and the data combined to ensure data collection in case of instrument failure (Section 2.1.2). Finally,

Site 2 is the tall tower measurement site at Tacolneston which has inlets at three heights, 54 m, 100 m and 185 m above the ground. This study uses an average of the 54 m and 100 m observations as a method to reduce local source influences. Differences in inlet altitude amongst the observation sites were represented in the atmospheric dispersion model (Section 2.2.1).



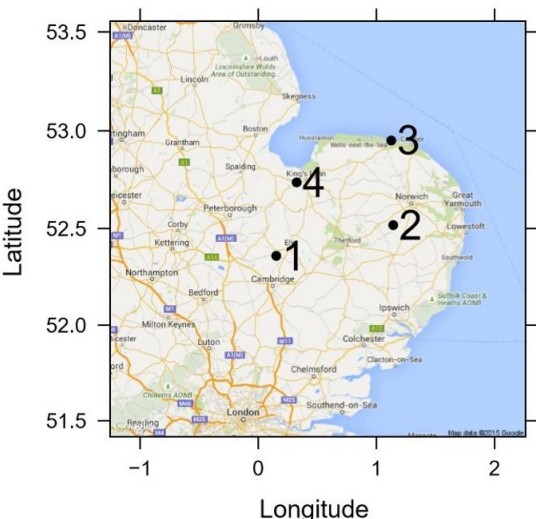

**Figure 1: Map of the East of England showing site locations. 1=Haddenham, 2=Tacolneston, 3=Weybourne, 4= Tilney. (Google Maps, 2015). See also Figure 1 in Palmer et al. (2018) for location of all UK monitoring sites, which are used in Section 3.5 in this paper.**

**Table 1: Overview table of the East Anglian measurement site information.**

| Site | Site name | Latitude, Longitude | Inlet height (m agl) | Instrument | Running dates |
|---|---|---|---|---|---|
| 1 | Holy Trinity Church, Haddenham | 52.359, 0.149 | 25 | GC-FID | 06/2012-Present |
| 2 | Tacolneston tall tower | 52.518, 1.139 | 54, 100 | Picarro CRDS | 07/2012-Present |
| 3 | Weybourne | 52.950, 1.122 | 10 | GC-FID UCAM | 02/2013-05/2014 |
| | | | | GC-FID UEA | 03/2013-05/2018 |
| 4 | All Saints Church, Tilney | 52.737, 0.321 | 25 | GC-FID | 06/2013-Present |

### 2.1.2 Instrumentation

The Tacolneston measurements were made using G2301 (Picarro Inc., USA) Cavity Ring-Down Spectrometer (Crosson, 2008) in the set-up described in Stanley et al. (2018). All other locations used gas chromatography coupled with flame ionisation detectors (GC-FIDs). At these locations, a stainless steel mesh (2 μm) was fitted to the inlet tube to filter any larger impurities from damaging the GC and reducing the air flow. The Weybourne site hosted two GC-FID instruments that shared the same inlet tube. The setup below describes the GC-FID installed by University of Cambridge (GC-FID UCAM). The setup of the second GC-FID, maintained by the University of East Anglia (GC-FID UEA), can be found in Forster (2013). The two data sets are combined in this project, noting differences in sampling intervals (1-2 minutes UCAM, ~20 minutes UEA).

A schematic of the GC-FID instrumental setup used at Sites 1, 3 (UCAM only) and 4 is shown in Figure 2. Nitrogen was used as a carrier gas. Two other gases, pressurised air and hydrogen, were used to fuel the flame within the FID. All three gases first passed through a molecular sieve to filter out water and hydrocarbons (labelled W and HC). The nitrogen carrier gas was additionally scrubbed for oxygen to protect the column from oxidation (labelled O). Inlet and calibration





tubes were filtered using a desiccant-based Nafion dryer (labelled ND). The GC was run at an internal temperature of 100°C and a column pressure of 34 psig. This setup allowed for a fast methane elution time (< 1 minute).

Samples were taken every 1-2 minutes at all GC-FID sites. The raw data were analysed using the commercially available software 'Igor Pro' (WaveMetrics, 2012), which automatically detects and measures the desired peak height. Relative

standard deviation (precision) values were defined half hourly. Average values showed precision to be below 0.3% (of the relative standard deviations).

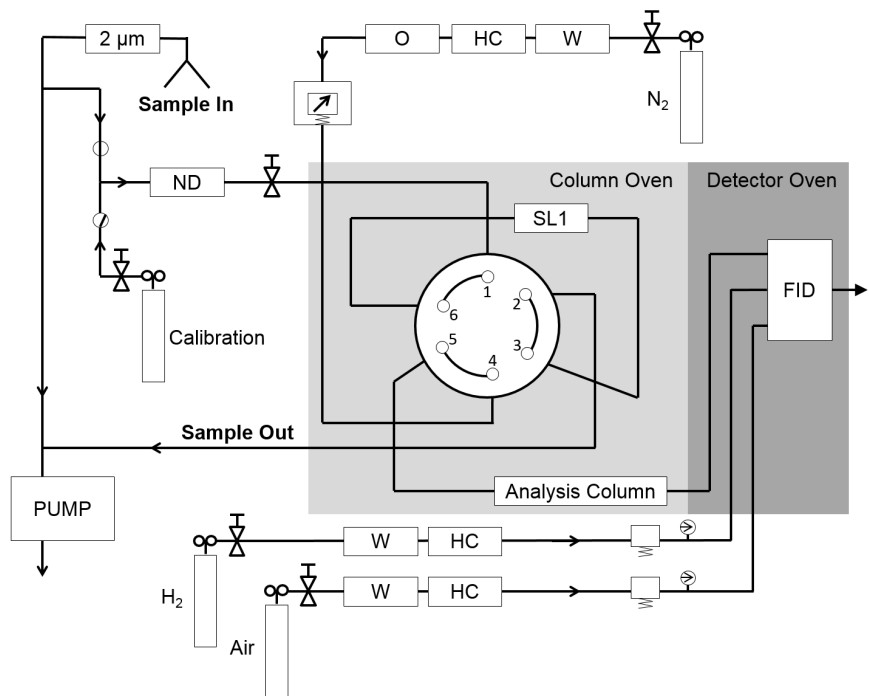

**Figure 2: Ellutia GC-FID 200 Series system flow diagram. Nitrogen carrier gas is filtered for water (W), hydrocarbons (HC),**
**and oxygen (O) before entering the column. Hydrogen and compressed air are used to fuel the flame ionisation detector (FID). Both are filtered for W and HC. A funnel filled with stainless steel-mesh (2 µm) is attached to the inlet tube, which is faced down to protect from rain and large particulates. Solenoid valves allow the GC to sample either the inlet air or the calibration gas. A pump is attached to draw in the inlet air and draw out the sample from the GC-FID. NB: SL1 = sample loop 1. ND = Nafion dryer.**

### 2.1.3 Calibration

All GC-FID sites were calibrated using an NPL calibration gas (0.28% precision). Inter-calibration experiments between our NPL-calibrated instruments (Site 1, GC-FID UCAM at Site 3, Site 4) and the NOAA-calibrated instruments (Site 2, GC-FID UEA at Site 3) showed an offset of -4.9 ppb (average of three calibration experiments). Although both the stated

and derived calibration concentrations for the NPL standard were within the ranges of the calibration gas uncertainties plus GC-FID precision, all NPL measurements were converted to the NOAA scale, consistent with the DECC network (Stanley et al. 2018).





## 2.2 Inversion Modelling

### 2.2.1 NAME trajectory calculation

The UK Met Office's Numerical Atmospheric dispersion Modelling Environment (NAME) model (Jones et al., 2007) is used to estimate air flow from potential methane sources to the measurement sites. NAME was originally developed by

the UK Met Office for modelling the long-range dispersion of radioactive material from nuclear power stations (Maryon et al., 1991). It is a Lagrangian model which uses the 3-D meteorological fields produced by the UK Met Office's numerical weather prediction model, the Unified Model (UM; Cullen 1993). When run backwards in time, NAME dispersion trajectories are used in the inverse modelling of atmospheric emissions (Ashfold et al., 2014; Manning et al., 2003). This project used two resolutions of UM meteorological fields: global (3 hourly, ~25 km horizontal, 8 levels in

the lowest 500 m vertical[1]) and UK (hourly, ~1.5 km horizontal, 12 levels in the lowest 500m vertical), the 1.5 km UK fields were nested within the global data when running NAME.

The model setup at each site is identical except for the particle-source release location (latitude, longitude) and height (m above ground level). The three sites with inlets between 15-27 m (1-Haddenham, 3-Weybourne and 4-Tilney) have a modelled release altitude of 25 m (±25 m) above sea level. Tacolneston (Site 2), with inlets at 54 m and 100 m is assigned

a release altitude of 75 m (±25 m).

NAME produces a modelled representation of the contributing 'surface influence' (defined as the lowest 100 m above ground level in NAME) at a particular location (one of the measurement sites) by releasing chemically inert particles (10,000 h$^{-1}$) from the $x, y, z$ coordinate of that measurement site. NAME computes the movements and geolocation of each particle every minute for 5 days backwards in time. Each location releases mass at a rate of 1 g s$^{-1}$ equally distributed

across the particles. A time integrated particle density map (units g s m$^{-3}$; resolution 1.5 × 1.5 km) is produced for each measurement location that shows, on a gridded output, the relative contribution that each grid square has made over the preceding 5-day period (Manning et al., 2011). After conversion, the resulting metric (units of s m$^{-1}$) can be described as the mean time that particles reside in each grid cell for a 1 hour particle release period. This metric corresponds to the multiplying factor by which emissions are diluted from their initial source to being monitored at the measurement location.

This relationship is given in Equation 1 and a dilution map for the measurement site Haddenham is shown in Figure 3. Dilution maps are calculated hourly at each measurement location over the monitoring period (of two years). Results are compiled into a 'dilution matrix', which shows how the dilution values changed over time, and is an input into the inversion system.

The model domain limits, shown in Figure 3, are centred on East Anglia but span most of the south east of England. With

the model particle lifetime set at five days, this is long enough for the vast majority of particles to leave the domain of interest, and thus capture all surface influence within the geographical domain.

---

[1] UM vertical resolution levels decrease as the altitude increases.




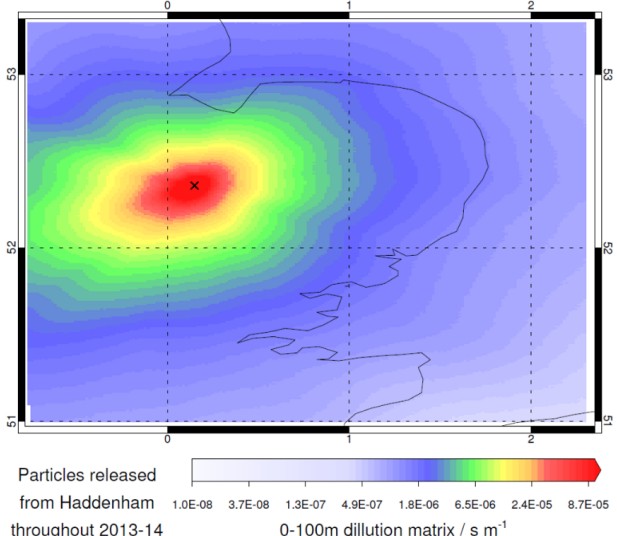

**Figure 3: Mean dilution matrix for the Haddenham site (Site 1, location marked with an X) for 2013 and 2014, as calculated by the NAME particle dispersion model. 3-D meteorological fields produced by the UK Met Office's numerical weather prediction model, at 1.5 km regional resolution nested within 25 km global resolution are used when running NAME. Sources are released for a one hour duration period, every hour from 01 January 2013 to 31 December 2014. Particles' geolocation is calculated every minute for 5 days backwards in time. Each location releases particles at a rate of 1 g s$^{-1}$, resulting in a time integrated particle density map (units g s m$^{-3}$; resolution 1.5 × 1.5 km) that shows the relative contribution of each grid square over the preceding 5-day period (Manning et al., 2011). A conversion to the 'dilution matrix' (units of s m$^{-1}$) can be described as the mean time that particles reside in each grid cell given a 1 hour particle release period.**

### 2.2.2 InTEM$_{2014}$ inversion model

InTEM$_{2014}$ uses methane measurements and the corresponding dilution maps to estimate emissions within a given domain according to the relationship expressed in Equation 1.

$$\text{emission (g s}^{-1}\text{ m}^{-2}\text{) x dilution (s m}^{-1}\text{) = concentration (g m}^{-3}\text{)} \tag{1}$$

Through an iterative process known as simulated annealing (Manning et al., 2003), pseudo-observations calculated from simulated emissions fields for specific times and locations are quantitatively compared with the measured observations using cost functional analysis. The resulting InTEM$_{2014}$ emissions estimate will be the emissions field with the lowest cost score.

Pseudo-observations are calculated by multiplying the emission estimates (from the 2012 NAEI) for each grid cell by their corresponding dilution value (taken from NAME) at each timestep. A least-squares cost function used (Equation 2) to quantitatively compare the two observational time series, similar to cost functions used in work such as Manning et al. (2003), Ashfold et al. (2014), and (Fang et al., 2016). Bayesian approaches have become more regularly used in top-down emission estimates, which incorporate pre-defined uncertainty estimates that result in calculated uncertainties for the final emissions (e.g., Arnold et al., 2018; Bousquet et al., 2011; Feng et al., 2018). Hierarchical Bayesian cost functions have since been developed where both the uncertainty values associated with the prior estimates, and the model uncertainty estimates can be derived within the inversion itself (Ganesan et al., 2014; Lunt et al., 2016). Estimating realistic and rigorously derived emission uncertainties remains a major challenge in inversion studies.





The cost function used here incorporates defined uncertainties associated with the observations and the model but does not include the use of a prior (see Section 2.2.3).

$$r_i = \sum_{i=1}^{n} \left( \frac{(y_i - (Kx_i)^2)}{(\sigma_\varepsilon)_i^2} \right) \tag{2}$$

$K_i$ is the forward model and $x_i$ is the measured concentration at a particular timestep ($i$). At all timesteps the difference between the pseudo-observation ($Kx_i$) and the measured observation ($y_i$) is squared and then divided by the uncertainty variance $(\sigma_\varepsilon^2)_i$. This uncertainty is the sum of all assumed errors in observations, modelling and baselines for each hourly timestep ($i$). Observational uncertainty is defined as the sum of the hourly instrument precision, the calibration gas uncertainty, and the standard deviations of the hourly concentrations plus 5 ppb. The value of 5 ppb is an estimate of the uncertainty in the baseline value in any given hour. Dividing by the total uncertainty (i.e., variance) de-weights uncertain observations. The lower the resulting cost score the smaller the difference between the pseudo and measured observations, implying a more accurate emissions estimate than one with a higher cost score.

The simulated annealing method in InTEM$_{2014}$ iteratively converges on the best solution and the final result is limited by the available computer resource. Therefore, the derived final output is close to but may not be the best possible solution, i.e., with the lowest possible cost score. For this reason, and due to the stochastic nature of the convergence within the simulated annealing process, the InTEM$_{2014}$ runs were repeated multiple times and the resulting emission results averaged. Sensitivity analyses showed that 25 repeats were sufficient to produce consistent methane emission estimates, standard deviations and cost scores.

### 2.2.3 A priori emission estimates

Unlike Bayesian inversions, InTEM$_{2014}$ does not use a prescribed *a priori* emission estimate, which includes boundaries to the emission magnitudes based on uncertainty assumptions. In this study, we used a random, non-negative emission field, which assumes no a priori knowledge at the location of emissions.

### 2.2.4 Solution grid

The methane emission estimates are resolved on a more spatially coarse grid than the NAME model output to reduce the computational cost and decrease modelling uncertainties (Manning et al., 2003). This so-called 'solution grid', of which the a posteriori estimate is resolved, is irregular and is constructed using the dilution matrix (Section 2.2.1) and the 2012 NAEI for methane, but the NAME output grid can first be divided into broader regions to calculate emission totals. These regions are based on the East Anglia county boundaries, providing rough county-wide estimates of methane emitted over the given period of time. The solution grid resolution is a sub-division of these county areas, which has a spatial resolution that is between the county-based starting regions and the NAME grid. The solution grid resolution is determined through two factors: the dilution matrix and the NAEI methane emissions values.

For the dilution grid matrix, areas where trajectories spent relatively long periods of time will have a finer spatial resolution, as more data are available and thus there is greater sensitivity to resolve the emissions from that area. A pre-defined dilution threshold subdivides regions into finer grids based on the dilution matrix (Manning et al., 2011). Generally, areas nearer to the measurement sites are more finely resolved than more distant areas. Complementing this, the NAEI methane emissions are also incorporated to define the solution grid resolution. The NAEI emission magnitudes are used as a linear scaling factor to define the grid resolution (as in other inversion techniques such as Rigby et al. (2011).





High methane emission sources in the NAEI (e.g., landfills) are more finely resolved than low methane emission areas. The resulting solution grid resolution can be seen in Figure 4.

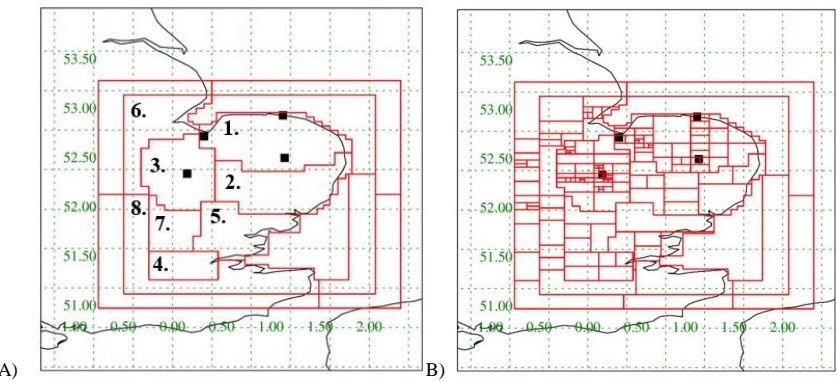

**Figure 4: A) Map showing the starting resolution of the inversion. These regions are loosely based on the county regions in East Anglia. As an output, InTEM provides statistical information on the emissions for each of these regions. B) Map showing the solution grid resolution. This grid resolution is computed based on information from the dilution matrix (Figure 3) and the 2012 NAEI for methane (Brown et al. 2018). Numbering refers to counties or other more arbitrary areas: 1= Norfolk; 2= Suffolk; 3= Cambridgeshire; 4 = London; 5 = Essex; 6 = Lincolnshire; 7 = Buckinghamshire; 8 = South west area.**

### 2.2.5 Baseline

A baseline that represents the atmospheric methane concentration arriving at the edge of the inversion domain (Figure 4) must be defined within InTEM$_{2014}$. For this study, a statistical baseline was calculated from the measured observations that also incorporated the particle trajectory analysis from NAME. Methane concentrations from the four measurement sites were divided into time series depending on whether their trajectory origin was dominated by a certain direction (e.g., from the NNE, ENE, ESE, NNW etc.). These eight individual time series, representing concentrations from the eight different compass directions, were used to estimate eight statistically derived baselines, each calculated by passing a rolling 18$^{th}$ percentile, spanning one week, through each dataset. The 18$^{th}$ percentile is chosen from a sensitivity analysis, in which the rolling percentile was varied from the 5$^{th}$ to the 45$^{th}$ percentile. The 18$^{th}$ percentile produces emission results with consistently stable emissions and with the lowest cost score of all the baselines tested. Baselines for the four measurement sites were then created using the NAME trajectory analysis to weight a combination of the eight direction dependent baselines.

### 3. Results

This section covers results from the InTEM$_{2014}$ inversion analysis. For analysis of the concentrations from the measurement sites, including their daily, weekly and intra-annual variability, please refer to Figure 3 in Palmer et al. (2018).

### 3.2 InTEM$_{2014}$ emission estimate results

The following results were derived using a one year dataset from all four sites covering the period from June 2013 to May 2014 (inclusive) with hourly observations. This period was chosen as it marks the first full year where all four sites were operational. A subset of the resulting measured and pseudo-observations for 01-30 July 2013 is shown in Figure 5. The equivalent 2012 NAEI pseudo-observations are included for comparison. From the measured observations, it is clear that



the Haddenham and Tilney sites observe short periods of elevated methane, usually during nocturnal hours when the boundary layer height is low, which suggests the presence of local methane sources. Several landfill sites can be found close to Haddenham and Tilney (<10 km) which can be large point sources of methane (NAEI, Brown et al., 2018). Isotopic analysis by Riddick et al. (2017), confirmed a methane signal from the Waterbeach Waste Management Park being present at the Haddenham site at times of elevated methane.

Figure 5 shows a large fraction of the *a posteriori* estimates lie outside the uncertainty range of the measured observations, although without a more thorough description of a priori errors this is difficult to fully diagnose. As it stands, this implies that either the prescribed InTEM$_{2014}$ uncertainties are too small or the resulting emission field needs be more resolved (in time and space) in order to better represent the concentrations being measured (currently, the spatial resolution of the resulting emission grid could be too coarse to fully capture the peaks and troughs of the measured time series). It should be noted that the pseudo-observations calculated using the NAEI are substantially outside the observation uncertainty ranges, and that neither are able to replicate the high concentrations measured at the Haddenham and Tilney sites. A scatterplot of a posteriori enhancements vs. observed enhancements as calculated by InTEM$_{2014}$ at the Haddenham site can be found in Riddick et al 2017 (Figure SM2.2). The InTEM$_{2014}$ resulting emissions field has a lower (i.e., better) cost score than the emissions grid from the NAEI (12.5 compared to 14.9). For comparison, the 2009 NAEI emissions grid yielded a higher cost score of 15.8, showing that the methane emissions distribution produced by InTEM$_{2014}$ fits the measured observations better than the 2009 or the 2012 NAEI.

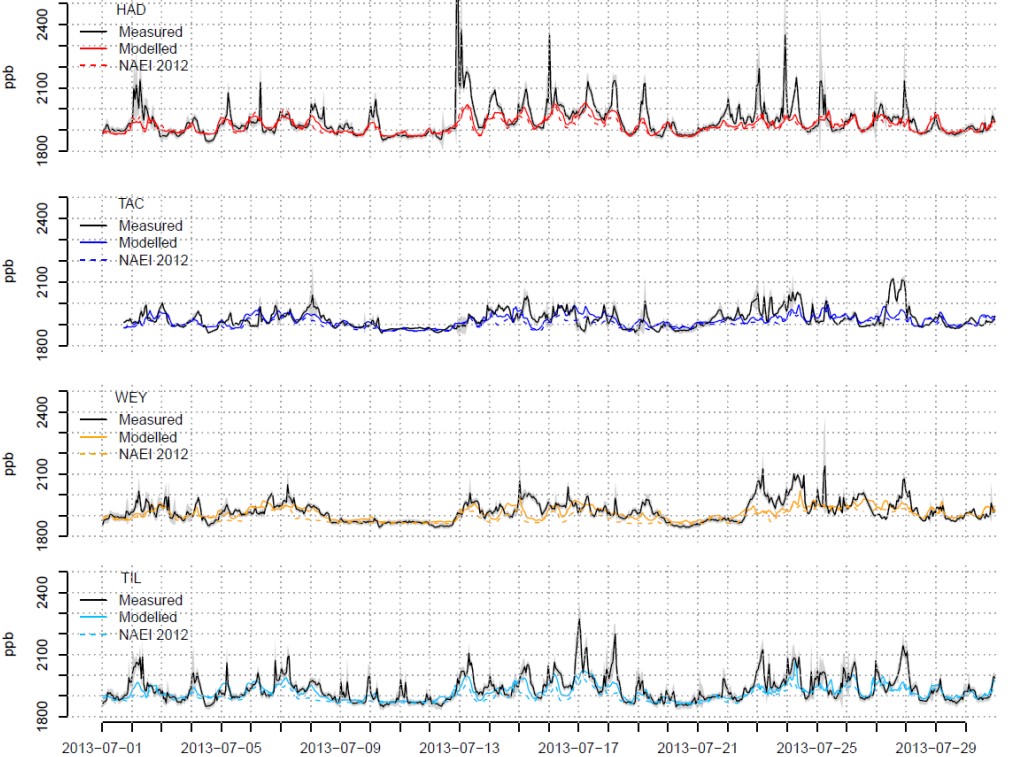

**Figure 5: Time series of measured and *a posteriori* modelled methane (using inversion estimated emission distribution) mole fractions for all 4 observations sites for 01-30 July 2013. The equivalent pseudo-observations calculated using the 2012 NAEI emissions inventory (Brown et al. 2018) have been added for comparison.**





### 3.2.1 Regionality

Figure 6 shows the inversion emission map and the 2012 NAEI methane emission map, both plotted to the same spatial resolution (see Section 2.2.4: Solution grid). The inversion produces comparable emissions estimates for East Anglia with total estimates being within 15% of the NAEI (NAEI estimates 280 kt yr$^{-1}$, InTEM$_{2014}$ estimates 310 ±63.0 kt yr$^{-1}$, rounded

to 2 s.f.). Similarities between the spatiality of emission are visible, with both maps showing large emissions in the London area, point sources around Haddenham, and lower emissions along the southern East Anglian coast. Discrepancies appear between some of the magnitudes in the finely resolved emissions maps, but local studies using additional measurements, Gaussian plume and WindTrax modelling do show that the high point source emissions near Haddenham are real (Riddick et al., 2017).

Table 2 shows the inversion area emission totals (labelled in Figure 4). The East Anglian areas are loosely based on the UK counties (Suffolk - 4, Norfolk - 10 and Cambridgeshire - 15). A positive relationship between area standard deviations and the distance from the measurement sites can be seen in Table 2. For example, the areas close to London and in the south west of the regions have standard deviations of 18.0 and 59.2, respectively, but areas representing the east Anglian counties are all below a standard deviation of 2.5. This implies that InTEM$_{2014}$ is able to more robustly resolve emission

totals for areas close to measurement sites, although individual site biases apply (see Section 3.4). This analysis implies that the ~15-25 m a.g.l. EA measurement sites have effective footprints of roughly a 50 km radius. Our estimates for methane emissions from Norfolk and Suffolk show good agreement with the estimates in NAEI, with differences of ~5%. Larger differences are found for Cambridgeshire where our estimate is 22.5% lower than that of NAEI. Percentage differences for regions that are further away from the measurement sites range from 10.8% (region 11, London area) to

66.1% (region 1, south west area). All land area estimates are within a factor of two of the NAEI.

Compared to the NAEI, InTEM$_{2014}$ emissions have a 'dipole effect' in some areas. For example, a large methane source is shown to the south west of Tacolneston but low emissions are estimated in the surrounding area. The NAEI also shows an increased emission level south west of Tacolneston, but the overall emission ranges are less extreme. It is unclear if these dipoles are 'true' signals, or a product of InTEM$_{2014}$'s inability to fully resolve emissions on this spatial and temporal

scale. Differentiating between false dipoles and real point sources is not straightforward in this analysis. Intermittent source emissions, as well as uncertainty in the meteorological analyses used to run NAME could account for InTEM$_{2014}$ being unable to pinpoint some emission sources. In principle, this could be overcome by introducing point sources into the priori, as used in some Bayesian approaches (Rigby et al., 2017).






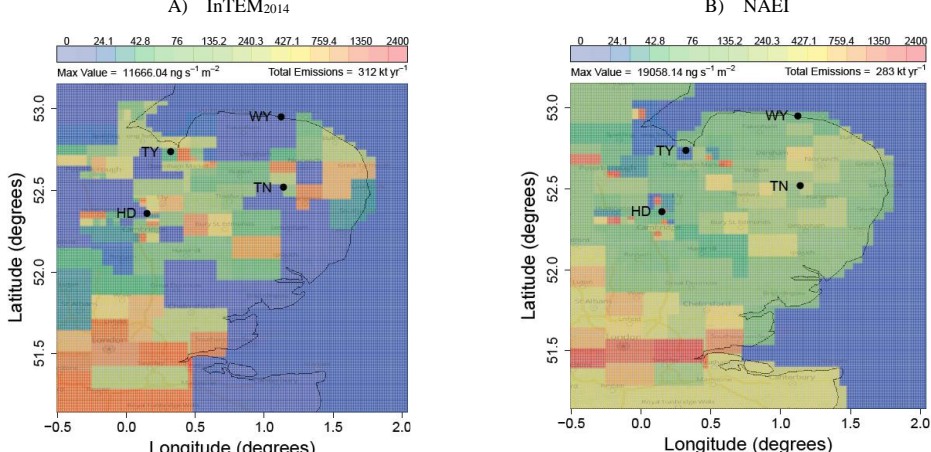

**Figure 6: A) Methane emission map, from June 2013 to May 2014, produced by an InTEM inversion run using all 4 sites' observational data. B) The 2012 NAEI (Brown et al. 2018) re-gridded to the inversion grid resolution (see Figure 5). Sites are labelled for reference: HD = Haddenham, TN = Tacolneston, WY = Weybourne, TY = Tilney. NB: Logarithmic colour scale. Difference between orange / red is roughly a factor of 100 larger than the difference between blue / green.**

**Table 2: Emission totals (kt yr⁻¹) resulting from InTEM inversion using all 4 observational site data for the period June 2013 to May 2014. Emission totals are for 'regions' shown in Figure 5. Equivalent totals of the 2012 NAEI (Brown et al. 2018) per region and their differences as percentages are shown for comparison. One standard deviation (1.s.d.) is shown below regional estimates. Sea regions from Figure 5 not shown here but emissions totals were 0.4 kt yr⁻¹ (0.64 1.s.d.) compared to 0.7 kt yr⁻¹**

**from the NAEI (57.1% difference).**

| # | | NAEI | InTEM ± 1 standard deviation | % difference |
|---|---|---|---|---|
| 1 | Norfolk | 38.9 | 37.1 ±1.7 | 4.7 |
| 2 | Suffolk | 24.1 | 22.8 ±1.9 | 5.6 |
| 3 | Cambridgeshire | 26.5 | 20.5 ±2.1 | 22.5 |
| 4 | London | 51.2 | 45.7 ±18.0 | 10.8 |
| 5 | Essex | 24.5 | 19.6 ±8.1 | 19.9 |
| 6 | Lincolnshire | 17.6 | 9.1 ±4.1 | 48.3 |
| 7 | Buckinghamshire | 20.5 | 30.6 ±7.3 | -49.1 |
| 8 | South west area | 75.0 | 124.5 ±59.2 | -66.1 |
| | **TOTAL** | 278.3 | 310.5 ±63.0 | -11.4 |

**3.2.2 Methane emission estimates surrounding Haddenham**

Due to how the spatial resolution of the emission grid is calculated (Section 2.3.3), finer spatial resolution is available in areas around the measurement sites. Figure 7 shows a magnified section of the emission maps in Figure 6 centred around the Haddenham site (Site 1) from (a) the inversion study and (b) the NAEI. Although, the uncertainty associated with

point sources is high (inversion standard deviations can be ~100% or larger for individual point sources, (Riddick et al., 2017), interesting aspects are discernible from the existence and locations of some of these emissions. Firstly, all point sources in the NAEI (Figure 7b) correspond to landfill sites, with the exception of the most southerly point source, which is the city of Cambridge. The inversion resolved these emissions, although all emissions west of Haddenham are lower than the NAEI. Furthermore, our analysis finds fewer emissions in the area labelled "1" compared to the NAEI. This area

corresponds to 'historic' landfills that are no longer in use (decommissioned in the late 1980s / early 1990s, Environment Agency, (Anon, 2015) yet are still estimated to be emitting methane in the NAEI. Hegde et al. (2003) investigated methane




emissions from a landfill in Taiwan and observed that buried waste had a peak emission between two and three years after burial and that emissions after five years were 0.63% of the maximum values measured. This analysis suggests emissions to be lower than calculated in the NAEI, although model uncertainties are significant. The area labelled "2" in Figure 7 shows another discernible difference between the inversion and NAEI emissions, with the inversion results

showing larger methane emissions. This area corresponds to managed and unmanaged fenland with multiple irrigation channels structured throughout. Areas of near-stagnant water can potentially be large methane emitters (Minkkinen and Laine, 2006) but few methane sources are estimated in the NAEI in this region (Brown et al., 2018). With this InTEM$_{2014}$ setup, our results suggest a potentially missing, or underestimated methane source in the NAEI for this area, although a more quantified uncertainty analysis would be needed as part of further work to resolve these emissions more fully.

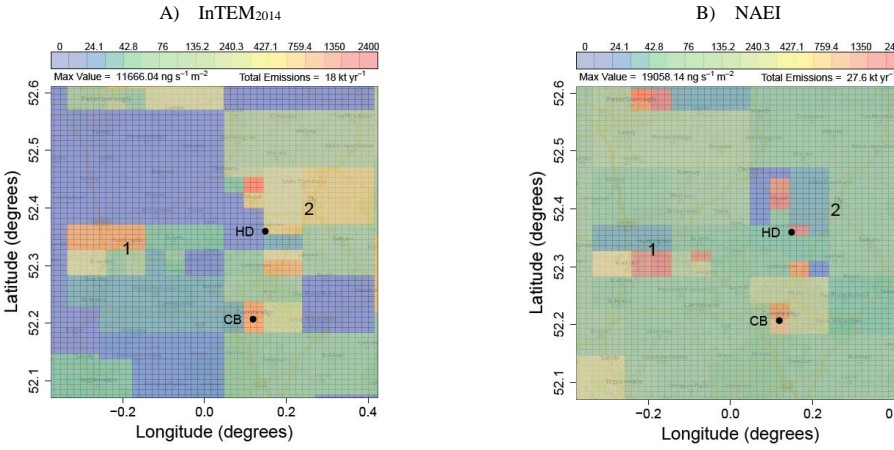

**Figure 7: A) Methane emission map, from June 2013 to May 2014, produced by the inversion using all 4 sites' observational data, zoomed to the Cambridgeshire area. B) The 2012 NAEI methane emissions (Brown et al. 2018) re-gridded to the inversion grid. Cambridge (CB) and Haddenham (HD) are labelled for reference. Label 1 refers to an area of active landfills. Label 2 refers to an area of manufactured irrigation channels, where stagnant water can accumulate.**

**3.4 InTEM$_{2014}$ sensitivity to the number of observation sites**

The final part of our analysis investigates InTEM$_{2014}$'s sensitivity to the number of measurement sites used within the inversion. For this analysis InTEM$_{2014}$ was run as described in Section 2.2 (one year period, June 2013 - May 2014) but the inversion was repeated using observation data from a subset of 1-3 measurement sites (all combinations were assessed). The InTEM$_{2014}$ emission estimates for the Norfolk, Suffolk and Cambridgeshire areas (areas 4, 10, 15) are

plotted in Figure 8 (referred to as NSC). This figure shows the range of NSC emissions totals is reduced as more measurement sites are incorporated in the inversion run. Furthermore, we can see that the inversion method is influenced by the specific sites' measurement data. For example, the sites which experience the lowest range of methane concentrations (Weybourne, Tacolneston) produce lower emission estimates for the NSC region (see Figure 5, and Figure 3 in Palmer et al. 2018). Similarly, sites with more local point sources (Haddenham, Tilney) produce higher regional

emissions maps.

NSC total estimates using a single measurement site in the inversion are further away from the NAEI but closer when all four measurement sites are used. However, in each inversion result using only one measurement site, the county estimates for the county that the single site resides compares more closely to the NAEI than other county estimates.





Poorly resolved local influences are diminished with the incorporation of other sites' data but not removed entirely. For example, inversions using Haddenham data always produces the higher NSC total emission estimates. These results strengthen the argument for incorporating multiple sites within inversion analysis but the number of required sites is dependent on the size and resolution, both spatial and temporal, of the desired region for analysis.

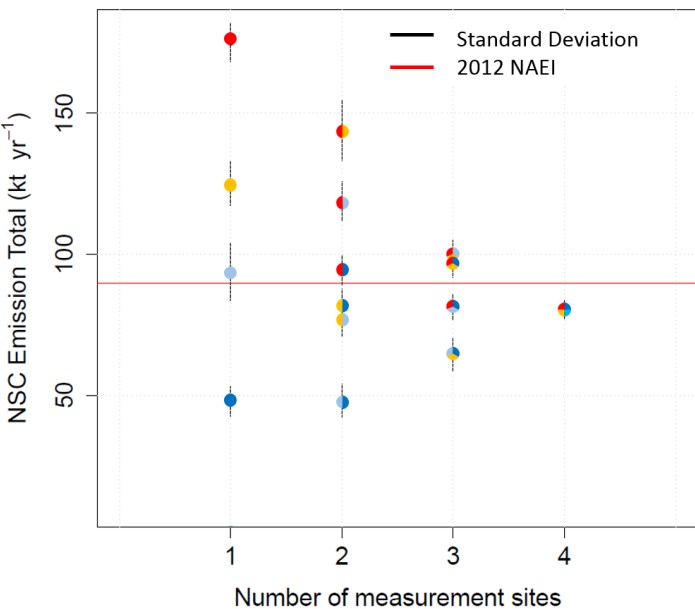

**Figure 8: Emission totals (kt yr$^{-1}$) for the three areas approximately corresponding to Norfolk, Suffolk, and Cambridgeshire (NSC), as shown in Figure 4. Emission result from InTEM inversions being run with 1-4 observational site(s) data. *x*-axis shows number of observation sites used in each InTEM inversion. Vertical black lines represent one standard deviation. Horizontal**
**red line shows NAEI emissions total for the NSC area. Colours shown on each dot correspond as Haddenham (red), Tacolneston (dark blue), Weybourne (light blue) and Tilney (orange). Colours also correspond to those used in Figure 5).**

### 3.5 Effect of including East Anglian sites in a national inversion

To investigate the influence of the East Anglia (EA) measurement network, InTEM$_{2018}$, as described in Arnold et al. (2018), was run both with and without the inclusion of the sites within the National UK network. Arnold et al., (2018)
uses an updated version of InTEM for estimating national emissions and the main differences with InTEM$_{2014}$ are summarised in Table 3.

Figure 9 shows the resulting methane emissions map centred over the east of England for A) the UK network including the EA measurement sites and B) without the EA network. Please note that Tacolneston is included in both inversions as it is a tall tower measurement site, however the measurement inlet heights vary for the different inversions (Table 3).
Table 4 show the InTEM$_{2018}$ estimated emission totals for an area closely corresponding to the counties Norfolk, Suffolk and Cambridgeshire, the three closest counties to the EA sites, calculated by InTEM$_{2014}$, both with and without the inclusion of the EA measurement sites. From this table, it is clear that the additional inclusion of the EA sites into InTEM$_{2018}$ does not greatly alter the emission estimates for the area but the uncertainty has been reduced (84.9 kT yr$^{-1}$, with 1.s.d. of 12.8, including the EA sites compared with 82.3 kT yr$^{-1}$ with 1.s.d. of 23.6 without). This is reassuring as it
implies robustness of the inversion results to additional data. Both InTEM$_{2018}$ estimates show a 45% increase compared to the 2015 NAEI. Additionally, the inclusion of the EA sites allows for finer spatial resolution to be resolved in the





national inversion, and thus provides further information. For example, Figure 9a shows a latitudinal band of larger methane emissions just north of Tacolneston (Site 2), a feature also visible in the InTEM sub-national inversions (Figure 6a).

In Table 4, the geographical boundaries between the InTEM$_{2014}$ and InTEM$_{2018}$ emission totals for the EA area vary slightly due to differences in the spatial resolution of the emissions grid, which result in the areas not being directly comparable. Nevertheless, a rough comparison shows similar totals, again demonstrating the stability of the inversion results, and both estimate higher emissions compared to the 2015 NAEI and lower compared to the 2012 NAEI.

**Table 3: differences in InTEM$_{2014}$ (Connors et al 2018) and InTEM$_{2018}$ (Arnold et al 2018). NB: a.g.l = above ground level.**

|  | *InTEM$_{2014}$* | *InTEM$_{2018}$* |
|---|---|---|
| *Observations:* |  |  |
| *Tacolneston measurement height* | Average of 54 and 100m | Average of 54, 100, and 185 m |
| *NAME dispersion model:* |  |  |
| *Definition of 'surface influence'* | lowest 100 m a.g.l | lowest 40 m a.g.l |
| *# of inert particles released* | 10 000 hr$^{-1}$ | 20 000 hr$^{-1}$ |
| *Particle tracking timestamps* | 5 days, every 1 minute | 30 days, every 1- 6 minutes |
| *Grid resolution* | ~1.5 km x 1.5 km | ~ 25 km x 25 km |
| *Inversion framework:* |  |  |
| *Prior* | none | 2015 NAEI (40% UK uncertainty) |
| *Cost function type* | Simulated annealing | Bayesian |
| *Solution grid resolution* | ≥4 km x 4 km | ≥25 km x 25 km |

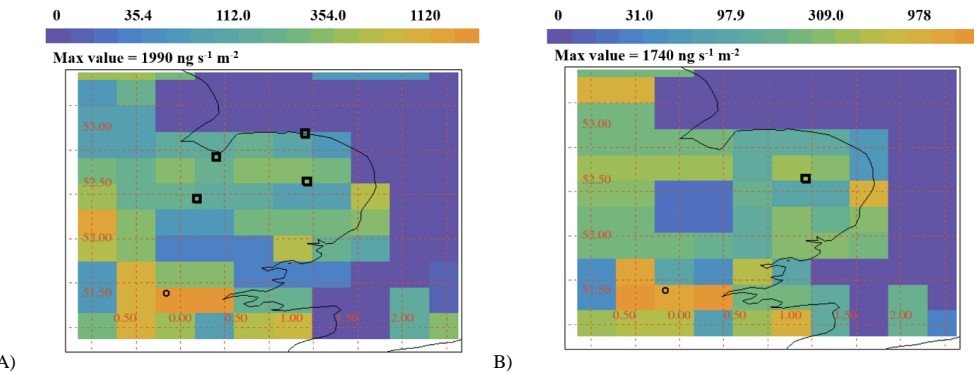

**Figure 9: InTEM$_{2018}$ emission results from the national inversion A) using Mace Head, tall towers, and East Anglian station data, and B) using just Mace Head and tall tower station data. NAEI 2015 emission estimates on a 25 km grid (UK assumed 40% uncertain) are used as prior for inversion.**





**Table 4: Comparison of methane emission totals for an area roughly corresponding to Norfolk, Suffolk and Cambridgeshire. NB: Due to differing spatial resolution of the emissions grid, areas are not equal and thus are not directly comparable. Additionally, different cost functions were used in the two inversion methods and the standard deviation from InTEM$_{2014}$ (marked with a *) is not directly comparable with InTEM$_{2018}$.**

| | InTEM$_{2014}$ | InTEM$_{2018}$ without EA sites | InTEM$_{2018}$ with EA sites |
|---|---|---|---|
| NSC area (kT yr$^{-1}$) | 80.4 | 82.3 | 84.9 |
| 1 standard deviation | 3.3* | 23.6 | 12.8 |

## 4. Summary and Discussion

We have employed a network of observations and an inversion system to estimate methane emissions over three counties in eastern England. This approach is conceptually similar to the ones used to estimate $N_2O$ emissions in the United States

mid-west (Nevison et al., 2018) and in California (Jeong et al., 2018) though ours is run on a smaller geographic scale and is trying to produce emissions at finer spatial resolution. To achieve this, measurements of methane from 4 sites in East Anglia were operated from 2012. The impact of local sources on the measurements was minimised by locating the inlets high in church towers in villages that are not part of the national gas distribution network.

These measurements were interpreted using a regional inversion approach based on the NAME inversion methodology

and high resolution Met Office 3-D meteorological analyses at 1.5 km x 1.5 km horizontal resolution nested within coarser analyses at 25 km x 25 km horizontal resolution. Baseline values were calculated using measurements on the upwind side of the area being studied. This approach produces emission estimates with fine spatial resolution (up to 4 km x 4 km). The resulting total emission estimates are in good overall agreement with the UK NAEI bottom-up estimates with several notable differences in the distribution of emissions. One difference is in the Fens region of East Anglia where we find

higher emissions. This could be due to emissions from managed wetlands currently being underestimated in the NAEI. The NAEI contains a number of point sources (such as landfills) whose presence can be clearly seen in the inversion analysis, even though using no emissions prior is used within the inversion. This is borne out in a case study examining methane emissions from the Waterbeach landfill site (Riddick et al., 2017). It implies that there is real spatial information in the inversion results, and that a more refined uncertainty analysis would allow emission estimates from point sources

to be derived from larger-scale analyses. Despite using a measurement-based approach to define the baseline, the level of knowledge of the methane concentration in the air entering East Anglia is a major cause of uncertainty in our analysis. Approaches in which East Anglia is nested within a larger scale inversion would be preferable (Manning et al., 2011). We have also investigated the impact of including the additional measurement sites in EA on calculated methane emission estimates in East Anglia using the national inversion approach InTEM$_{2018}$. Results from the inversion, which included the

national GHG network stations (UK DECC network, Stanley et al., 2018; GAUGE tall towers, Stavert et al., 2018) and the EA network, show consistent results to those just using the EA network, demonstrating a stability in the inversion 'top-down' estimates. Benefits of the addition of the EA sites within InTEM$_{2018}$ were the ability to provide finer spatial resolution and to decrease the associated uncertainty for that area.





**Author contributions**

S. Connors, A. J. Manning, and N. R. P. Harris designed the experiments and S. Connors, A. J. Manning, and A. D. Robinson, carried them out. S. Connors and A. J. Manning developed the model code and performed the simulations. A. D. Robinson, S. N. Riddick and R. L. Skelton sources, installed and ran the measurement instrument in Sites 1, 3 & 4 (Haddenham, Weybourne – UCAM instrument, and Tilney) and provided data for the analysis. G. L. Forster, D. E. Oram and S. Humphrey ran the measurement instrument in Site 3 (Weybourne – UEA instrument) and provided data for the analysis. Anita Ganesan, Aoife Grant, Kieran Stanley, and Ann Stavert, ran the measurement instrument in Site 2 (Tacolneston) as well as other data for the GAUGE network and provided data for the analysis. N. R. P. Harris and P. I. Palmer were project leads and gave scientific oversight and guidance throughout the planning, implementation, collection, and analysis of the data. S. Connors, A. J. Manning, and N. R. P. Harris prepared the manuscript with contributions from all co-authors.

**Acknowledgements**

This project was supported by the UK Natural Environment Research Council (NERC) through the Greenhouse gAs UK and Global Emissions (GAUGE) project on grant number NE/K002570/1. We also thank the Department of Environment, Farming and Rural Affairs and the Royal Society for seed funding and NERC for additional support through grants NE/G014655/1, NE/J006246/1 and a PhD studentship for Sarah Connors. We acknowledge the UK Government Department for Business, Energy & Industrial Strategy and the former UK Department for Energy and Climate Change for the use of the national methane measurement network data and we would like to thank the National Centre for Atmospheric Sciences (NCAS) for access to the Weybourne Atmospheric Observatory and for providing data. Special thanks to Holy Trinity church, Haddenham and All Saints Church, Tilney for allowing us to site our instruments in their churches.

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
