# Peer review of "Estimates of sub-national methane emissions from inversion modelling"

_Atmospheric Chemistry and Physics, 2018_

## Referee Comment (RC1) · Anonymous Referee #1 · 9 Jan 2019

**1 Overview:**

Review of "Estimates of sub-national methane emissions from inversion modelling" by Connors et al.

Connors *et al.* present an analysis from a year of methane measurements at 4 sites in East Anglia. They describe the 4 new instruments that are mounted on churches or other tall towers. These instruments are then used to constrain methane emissions in between June 2013 and May 2014. The description of the network is generally good and the figures are all reasonably well made. However the inversion portion of the manuscript needs quite a bit of work. I have some major concerns with the seemingly unsubstantiated choices, poor description, and potential overfitting. I think there is

interesting data here that should eventually be published, but there are some major issues that need to be dealt with first.

**2 Major Comments**

The section on inversion modelling (Section 2.2) could use a major re-write to describe what was actually done and justify choices made.

2.1 Over-fitting?

A **major** drawback with a least squares cost function is over-fitting. This is precisely why most inversions use a regularization or a prior. I don't see any discussion of over-fitting. How do the authors combat over-fitting?

I strongly suspect this is why they find a 'dipole effect' (see Minor Comment #1).

- 2.2 Simulated Annealing
- 2.2.1 Use of Least-Squares and Simulated Annealing?

A least squares cost function (also known as a maximum likelihood estimate), as the authors use, has a closed form solution for the optimal solution. Using the author's notation, the optimal solution  $(\hat{\mathbf{x}})$  would be:

$$\hat{\mathbf{x}} = \left(\mathbf{K}^T \Sigma_{\epsilon}^{-1} \mathbf{K}\right)^{-1} \mathbf{K}^T \mathbf{y}'.$$
(1)

where K is the dilution matrix,  $\Sigma_{\epsilon}$  is the error covariance matrix,  $\mathbf{y}'$  is the vector of observations after removing the background concentrations:  $\mathbf{y}' := \mathbf{y} - \mathbf{b}$ .
There are four cases I could envision using something like Simulated Annealing:

- 1. if the system is non-linear (e.g., if  $\mathbf{K}$  is a function of x)
- 2. to regularize the solution (e.g., by coarsening  ${\bf K}$  as part of the inversion)
- 3. there are additional constraints being applied (e.g., non-negativity)
- 4. for computational expediency

In the case of additional constraints (such as a non-negativity constraint), it seems that bounded optimization would be preferable to a stochastic method such as simulated annealing. Gradient-based methods like L-BFGS-B ("https://en.wikipedia.org/wiki/Limited-memory\_BFGS#L-BFGS-B") tend to be much faster for convex optimization problems such as this one. A stochastic method like Simulated Annealing would probably be better for a non-convex optimization problem.

**2.2.2 Error statistics from simulated annealing**

Regarding the use of Simulated Annealing, it's unclear to me why the authors chose to use a technique like simulated annealing here. Simulated annealing is an optimization technique that is quite efficient, so it works rather well in high-dimensional problems. However, the samples obtained from Simulated Annealing are inconsistent with the true posterior statistics (the uncertainties will be smaller that the true uncertainties). So reporting error statistics from Simulated Annealing strikes me as dubious at best.

Using something like an adaptive MCMC or a reversible jump MCMC (rjMCMC) as some of the co-authors here have previously done seems far superior to simulated annealing.
Related to this, it's entirely unclear why Manning *et al.* (2003) would be the citation for simulated annealing on Page 8, Line 26. This is not a technique or method that Manning developed. For example, the Machine Learning textbook from Christopher Bishop (Bishop, 2007) would be a much more appropriate citation for someone interested in how Simulated Annealing works.

**2.3 Section 2.2.3: A priori emission estimates**

It's not clear why this section is necessary at all. As mentioned above, if the system is linear (the dilution matrix is predetermined and does not change) and the authors choose a least-squares cost function then the solution can be directly computed. There would be no need for a "random, non-negative emission field". How is this being used? Is this the initial starting point for the simulated annealing? If so, then it should be referred to as such and probably not be discussed in a 2-sentence section on prior emissions.

**2.4 Regarding the resolution of the solution grid**

There has been an abundance of work looking at how to define a multi-scale state vector. The textbook from Rodgers (2000) talks about this. Bocquet *et al.* (2011; QJRMS) is entirely devoted to this topic. Other work like Turner & Jacob, (2015; ACP), Lunt *et al.*, (2016; GMD), Henne *et al.* (2016; ACP), and Bousserez & Henze, (2017; QJRMS) also talk at length about how to construct this multiscale formalism.

Briefly, by coarsening (or restricting) the grid you are applying a hard constraint on the inversion. Basically, sub-elements are no longer allowed to vary independently. This is a form of regularization. In the most extreme case you could coarsen to a single state vector element. The degree of coarsening can change the problem from an under-determined problem to an over-determined problem. This ties back to an earlier

**ACPD**
question, how do you deal with potential over-fitting? Was there any cross-validation done?

At the bare minimum, Bocquet et al. (2011) should be cited.

**2.5 Computing the Baseline**

This section needs more work. Computing the baseline is a non-trivial task for any regional inversion and many studies are devoted entirely to estimating the baseline. At the bare minimum, the authors should provide some justification for their choice of the  $18^{th}$  percentile. Other studies such as Henne *et al.* (2016; ACP) provided an extensive analysis on the choice of background including precomputing it (as Connors *et al.* have done) vs jointly estimating it as part of the inversion.

2.6 Evaluation of meteorology?

I didn't see any mention of evaluating the meteorology. This is a crucial step in atmospheric inversions using real measurements that seems to be missing.

**2.7 Figure 3 seems odd**

The dilution matrix as Connors *et al.* refer to it (also commonly known as the footprint matrix, transport operator, Jacobian, etc.) looks pretty Gaussian. Is there no dominant wind pattern at Haddenham? I would usually expect some dominant wind pattern (i.e., more sensitivity to the upwind region). A wind rose showing that, indeed, the winds are roughly uniformly represented from all sectors here would be useful. Otherwise, turning this into a 3 panel figure (current figure as a large column on the left and two subpanels on the right column) with two illustrative examples of dilution matrices from
two days. I would expect the illustrative days to show strong sensitivity upwind of the site for that day.

Basically, I'm curious if this was computed correctly.

**2.8 Agreement with NAEI:**

The authors mention good agreement with NAEI (to within  $\sim$ 5%). However, Figure 6 looks strikingly different. Is this because they've further coarsened the emissions before this comparison? I don't see how they are getting a 5

**3 Minor Comments**

3.1 Dipoles and overfitting in the solution

The authors discuss a 'dipole effect' (e.g., Page 11, Line 21) in the inversion results. These are common in solutions with overfitting. The inversion is attempting to fit a high value at a measurement site, so it inflates the emissions to a very large value at that one location and then compensates by reducing the emissions in a neighboring grid cell where the observations have weak constraints. Basically, this is what happens:

- Location A: concentration too low -> increase emissions in just this location
- Location B: no constraint on concentration, domain wide emissions too high because of Location A -> reduce emissions

This is combatted in most inversion systems by having a prior or regularization that includes some off-diagonal relationships (e.g., emissions from Location A and Location B should be weakly correlated).
It would be nice if the authors specified *where* the work was done in the title. I'd suggest adding "in East Anglia" or "in the United Kindgom" (or something to that effect). Maybe something like: *"Estimates of sub-national methane emissions from inversion modelling in East Anglia"*

**3.3 Proof of concept**

On Page 3, Line 1 the authors motivate the work as a "Proof of Concept" that inversion schemes can work at sub-national scales. Although this has been shown numerous times in the past. Examples include Scot Miller's 2013 PNAS paper for methane in the US, Stephan Henne's 2016 ACP paper for methane in Switzerland, work from Kathryn McKain on methane emissions in Boston, work from Ken Davis' group on urban inversion modelling in Indianapolis for  $CO_2$ , and work from Thomas Lauvaux on  $CO_2$  at urban scales. So I don't find a "Proof of Concept" to be a particularly compelling motivation.

The work is definitely interesting, but I don't think this should be a major motivation for the reader.

3.4 Table 1:

The instrument acronyms are rather confusing in Table 1. I would remove "UCAM" and "UEA" from the table and instead add a different column that lists the sampling rate. Alternatively, the authors could have footnotes under that table that explain the acronyms. As it stands, the reader needs to scan the text to try and figure out what the acronyms mean.
**3.5 "Pseudo-observations":**

I would avoid using the term "pseudo-observations" because it sounds like the authors are doing a synthetic-data study (i.e., an OSSE). "Simulated", "modelled", or "predicted" concentrations would be preferable.

3.6 Equation #1:

Do not present an equation in this form. Use this equation to introduce your nomenclature for later. Something like this would be preferable:

$$\mathbf{y} = \mathbf{K}\mathbf{x} + \mathbf{b} \tag{2}$$

where y is an  $n \times 1$  vector of concentrations (units: g m-3), K is an  $n \times m$  dilution matrix (units: s m-1), x is an  $m \times 1$  vector of gridded emissions (units: g s-1 m-2), and b is an  $n \times 1$  vector of concentrations upwind of the modelling domain (units: g m-3).

3.7 Table 3:

"Simulated annealing" is not a cost function type. The cost function is a least-squares or maximum likelihood estimate.

---

## Referee Comment (RC2) · Anonymous Referee #2 · 26 Feb 2019

The manuscript "Estimates of sub-national methane emissions from inversion modelling" by Connors and co-workers presents atmospheric inverse modelling results from a very dense methane surface observation network in the eastern UK, quantifying local to regional sources and investigating the impact of network density. In terms of spatial measurement density and targeted local point sources the study offers some new insights into the feasibility of inverse modelling at this scale. My main concern (details below)with the present study is the use of unfiltered data for the inverse modelling and the use of a less well controllable inversion system. Hence, I suggest some major revisions to the manuscript before it can be published in ACP. Otherwise the manuscript is well written and structured, but a number of figures need to be improved before publication as well (details below).

**Major comments**

**Use of unfiltered observation data in the inverse modelling**

The tree additional CH4 observing sites setup for this study have sampling heights between 10 and 25 m above ground. Although the authors indicate that local sources should be small (a notion that is not well documented) one can see considerable concentration increases at all 3 sites during nighttime stable boundary layer conditions (Fig 5). It is well known that atmospheric models face serious challenges in stable boundary layers. Hence, it is not surprising that there exists a large discrepancy between observations and simulations (both using NAEI emissions and optimised a posteriori emissions) for nighttime observations. The comparison during daytime is generally much better. The problem is less evident for the higher sampling height of Tacolneston. Other inverse modelling studies have often used filtered observational data, excluding nighttime stable conditions, to rule out that biases in the transport model impact the emission estimates. Why was this not done or at least explored in the current study? I strongly encourage to explore the impact of observation filtering on the emission results, especially also in the light of the large variability seen in the results when only sub-sets of sites were used.

**Inversion scheme**

It seems a bit of a shame that the authors chose to use the InTEM2014 inversion system for their main analysis, since the InTEM2018 Bayesian inversion system would have allowed for a more stringent analysis of uncertainties in the inversion. Since InTEM2014 does not prescribe a priori emissions and their uncertainties, it remains unclear how the final uncertainties in InTEM2014 were derived and how representative they are of the real uncertainties. In addition, a fully Bayesian approach could have dealt with what is described by the authors as 'dipole effect', by introducing spatial correlation in the a priori covariance matrix. Another problem of the annealing approach and the used cost function seems to be over-fitting of the data. Without a consistent uncertainty budget the method seems to be vulnerable to over-fitting and hence produces dipoles but possibly additionally noisy emission fields. Although the authors explain that InTEM2014 was ready to use in their group, it seem one could have learned a bit more with InTEM2018. Furthermore, some InTEM2018 results are presented in section 3.5 anyway. So why not use it for the main analysis as well?

**Minor comments**

p1,I33: What does 'similar methane estimates' mean in terms of percentage differences? Next sentence mentions 'good agreement' as about 5 %.

p2,I5: The sentence should mention when this rise started and that there was a decade of stabilised CH4 concentrations before. There is another recent publication that should be included in the list: Thompson, R. L., Nisbet, E. G., Pisso, I., Stohl, A., Blake, D., Dlugokencky, E. J., Helmig, D., and White, J. W. C.: Variability in Atmospheric Methane From Fossil Fuel and Microbial Sources Over the Last Three Decades, Geophys. Res. Lett., 45, 11,499-411,508, doi: 10.1029/2018GL078127, 2018.

p2,l9: Maybe one can say that emission reductions are feasible. However, more interesting would be a statement why such reductions are probably more feasible or easier to achieve than reducing CO2 emissions of a similar magnitude.

p2,I13: I would reformulate this sentence towards something that states that knowing what causes and where emissions occur allows to design efficient reduction strategies. Quantifying emissions with atmospheric observation offers independent validation/support tool to assess if reduction measures were successfully met.

p2,l24: Is the 40 % uncertainty of the NAEI on the 1-sigma or 2-sigma confidence level?

p2,I30: There are several other CH4 inverse modelling studies on the national or subnational scale. Not just the ones for the UK.

p3,l3: Why are only preliminary findings presented? Only final results should be pub-

**СЗ**

lished in a peer-reviewed manuscript! I suggest rewording.

p3,I22f: Why would that be the case? Also in an area with homogeneous emissions 4 sites at different locations and different meteorology would experience different concentrations and the challenge for the transport and inverse modelling system would largely be the same. What is more interesting from my perspective is the presence of different dominating emission sources in the domain. This allows the inversion system to pick up problems of the inventory in terms of biased emission factors for different processes. Something which is also discussed later by the authors.

p3,l32: What about other local sources typical for rural environments? Any livestock in these towns? Waste water treatments?

p3,I33: Any coastal wetlands or marshes nearby?

p4,I5: Can this number also be given as a mole fraction? 0.3 % of 1900 ppb? So in the order of 5 - 6 ppb?

section 2.1.3: How often was a calibration gas applied? Was this a one point calibration or multi-point calibration? Was an independent target gas used to estimate the performance of the calibration?

p6,I16f: How can the 100 m sampling height be justified at this scale? With a release at 100 m (Tacolneston) and a sampling height of 100 m one will get source sensitivity directly at the receptor, whereas in reality and especially during stable conditions I would expect a plume to take some time and distance before actually touching the ground. Usually this distance should be short compared to the grid scales of the atmospheric inversions, but here this grid scale is in the order of a few km only.

p6,I25 and elsewhere: Usually the term 'footprint' or 'sensitivity map' is used in this context. 'Dilution map' seems to suggest something else. It should also be mentioned that the figure presents average conditions whereas hourly 'footprints' are the ones that are used for the inversion.

Equation 1: Bit of a poor layout for an equation. Please use a more mathematical notation and explain units in text. There is also a sum over the domain required to yield the concentration at the receptor!

Equation 2: It remains unclear why one would need a simulated annealing method to solve for the minimum of equation 2. There should be a very straightforward analytical solution for this! Why does r have an index i if the sum runs over i? Does i run over all observations from all sites or just over time? Why does x have an index i? x is the state vector that does not change with time. K should have an index i or maybe one could write  $(Kx)_i$ . Also the text says that 'xi' is the measured concentration. That is wrong!

p8,l9f: Are individual uncertainty terms added directly or is a sum of squares used (which would be more appropriate)? What is the final average uncertainty? How does it differ for the different sites? This is important to understand if a given site has more influence on the results than others.

p9,118: What does stable emissions mean in this context? A posteriori emissions did not change with the choice of percentile threshold?

p9,118: How is the 'cost score of the baseline' derived? Is the baseline part of the state vector? Or does it remain unchanged?

p10,I15: What is the cost score? r in equation 2?

p11,I4: Give uncertainty estimate for NAEI value here as well. Somewhere it said +/-40 %. So 112 kt/yr? Or at least since the 40 % was given for the national total. Same question again: What is the confidence level of the uncertainties?

p11,l8: What is WindTrax modelling? Not clear if one does not want to read the reference. It would call it a local scale Lagrangian particle dispersion model.

p11,I8: "high point source emissions near Haddenham". Actually the large point sources around HD seem to be surprisingly well resolved by InTEM and NAEI. I am more concerned about the large emissions in InTEM east of TN. Could this be wrong

attributions from sources outside the UK (for example Benelux region)?

p11,I13: What are the units of the given standard deviations? Looks like these are values from Table2. So mass emissions? Giving a relative uncertainty would make more sense when comparing regions with strongly different total emissions.

p11,I14f: Unclear why a 'footprint radius' of 50 km would be implied.

Figure 6: It would also be interesting to see a difference map between InTEM results and NAEI inventory.

p12,I1: How comparable is a study of landfills in Taiwan with conditions in the UK? Environmental factors will play a large role in the decay processes in a landfill. These factors appear to be quite different between UK (temperate climate) and Taiwan (tropical). Also the question of how much and which kind of organic matter was initially deposited in the land fill, will play a role. Is there no similar study from a European site?

p12,I4f: This looks more like the emissions are less well allocated in InTEM compared to NAEI. The point sources east and north of HD are less intense in InTEM and may be wrongly allocated to the larger grid cells labeled 2. What are the total emissions for a region around the cells labeled 2 but including the point sources north and east of HD? I would expect that the total may be much more similar.

p13,I5: Are these irrigation or drainage channels? How are the managed fenlands used? Rangeland, crop agriculture? If the latter dominates it is likely that these lands are usually well drained and no large emitters of CH4. How large are the unmanaged fenlands in comparison? An inversion grid structure that would reflect different dominating land cover types could have helped to distinguish different source processes.

p13,I19: This does not fit to the area labels in Figure 4! Why were only these areas considered for the sensitivity analysis?

Section 3.4: The large impact/bias introduced by the Haddenham observations could

be a result of poor representation of the nighttime observations (see Fig. 5). I wonder if these results would be more robust if only daytime observations would be used. Also see major comment above.

Figure 9: Please use the same value range for both sub-panels in order to make them comparable. Alternatively, a difference plot would also emphasise the important details. Could the NSC area, as used for table 4, be outlined in the figures?

p16,I26: This (uncertainty of baseline) has not been discussed anywhere above. How do we know that this a major source of uncertainty? How is it quantified? Only an assumption is made for the baseline uncertainty (5 ppb) but there seems to be no justification of this value.

p16,I27: But that is exactly what was done in section 3.5. So why not use this as the main analysis using InTEM2018 instead of the older version?

p16,I32: 'finer resolution' This is not visible in Fig 9. which seems to show the same spatial resolution for both InTEM2018 runs, with our without EA sites.

**Technical comments**

Figure 1: Improve resolution of googleMaps image. Latitude and Longitude axis labels should also contain units.

Table 1: Units for longitude and latitude.

p5,I2: 'psig' not SI units.

Figure 6 and 7: The colours on the map are not very clear (line patterns). Looks like something went wrong during conversion to pdf.

---

## Author Comment (AC1) · 16 Apr 2019

Dear reviewer,

From myself and my co-authors, I would like to thank you very much for reviewing the manuscript. I attach the responses to your and the other reviewer's comments, directly followed by a track-changes version of the manuscript.

Best wishes, Sarah Connors

Please also note the supplement to this comment:
https://www.atmos-chem-phys-discuss.net/acp-2018-1187/acp-2018-1187-AC1-supplement.pdf

---

## Author Comment (AC2) · 16 Apr 2019

Dear reviewer,

From myself and my co-authors, I would like to thank you very much for reviewing the manuscript. I attach the responses to your and the other reviewer's comments, directly followed by a track-changes version of the manuscript.

Best wishes, Sarah Connors

Please also note the supplement to this comment: https://www.atmos-chem-phys-discuss.net/acp-2018-1187/acp-2018-1187-AC2-supplement.pdf

[Figure]

[Figure]

**Supplement:**

**Contents Table**

|   | Anonymous Referee #1                     | 2  |
|---|------------------------------------------|----|
|   | Received and published: 9 January 2019   | 2  |
|   | Anonymous Referee #2                     | 7  |
| 5 | Received and published: 26 February 2019 | 7  |
|   | Track-changes manuscript                 | 13 |

**Anonymous Referee #1**

**Received and published: 9 January 2019**

**1 Overview:**

10

5 Review of "Estimates of sub-national methane emissions from inversion modelling" by Connors et al.

Connors et al. present an analysis from a year of methane measurements at 4 sites in East Anglia. They describe the 4 new instruments that are mounted on churches or other tall towers. These instruments are then used to constrain methane emissions in between June 2013 and May 2014. The description of the network is generally good and the figures are all reasonably well made. However the inversion portion of the manuscript needs quite a bit of work. I have some major concerns with the seemingly unsubstantiated choices, poor description, and potential overfitting. I think there is interesting data here that should eventually be published, but there are some major issues that need to be dealt with first.

**2 Major Comments**

15 The section on inversion modelling (Section 2.2) could use a major re-write to describe what was actually done and justify choices made.

**2.1 Over-fitting?**

A **major** drawback with a least squares cost function is over-fitting. This is precisely why most inversions use a regularization or a prior. I don't see any discussion of overfitting. How do the authors combat over-fitting? I strongly suspect this is why they find a 'dipole effect' (see Minor Comment #1).

Response:

Thank you, we acknowledge the limitation of the least squares cost function. However there are also limitations with using a prior with high levels of unknown uncertainty, particularly at such fine spatial resolution on the sub-national scale. Therefore,

- 25 in this study we are assuming that the prior uncertainty is so high as not to add information to the inversion system. It is also important to recognize that if one uses prior information then the result is not independent of the prior and therefore cannot be used to verify the prior emissions, which is what we wanted to do in this study. Overfitting is definitely a potential hazard that is why the inversion was repeated with many different settings to explore the uncertainties in the system: these are discussed at length in Connors, PhD Thesis, 2015.
- 30 Please see Minor comment #1 response for discussion on the 'dipole effect'.

**2.2 Simulated Annealing**

2.2.1 Use of Least-Squares and Simulated Annealing?

A least squares cost function (also known as a maximum likelihood estimate), as the authors use, has a closed form solution 35 for the optimal solution. Using the author's notation, the optimal solution (^x) would be:

^x =□KT\_-1\_K \_-1KT yo. (1)

where K is the dilution matrix, \_\_ is the error covariance matrix,  $y_0$  is the vector of observations after removing the background concentrations:  $y_0 := y - b$ .

**40**

55

There are four cases I could envision using something like Simulated Annealing:

- 1. if the system is non-linear (e.g., if K is a function of x)
- 2. to regularize the solution (e.g., by coarsening K as part of the inversion)
- 3. there are additional constraints being applied (e.g., non-negativity)
- 45 4. for computational expediency

In the case of additional constraints (such as a non-negativity constraint), it seems that bounded optimization would be preferable to a stochastic method such as simulated annealing. Gradient-based methods like L-BFGS-B ("https://en.wikipedia.org/wiki/Limited-memory\_BFGS#L-BFGS-B") tend to be much faster for convex optimization problems

50 such as this one. A stochastic method like Simulated Annealing would probably be better for a non-convex optimization problem.

Response:

Thank you for this comment, we acknowledge that the simulated annealing method is not the fastest for convex optimization and we certainly do not want to claim that this is the most efficient solution method. At no point in the manuscript have we stated this. Simply, this is one method that can be used whilst imposing the constraint of a non-negative solution.

**2.2.2 Error statistics from simulated annealing**

Regarding the use of Simulated Annealing, it's unclear to me why the authors chose to use a technique like simulated annealing here. Simulated annealing is an optimization technique that is quite efficient, so it works rather well in high-dimensional

problems. However, the samples obtained from Simulated Annealing are inconsistent with the true posterior statistics (the uncertainties will be smaller that the true uncertainties).

So reporting error statistics from Simulated Annealing strikes me as dubious at best. Using something like an adaptive MCMC or a reversible jump MCMC (rjMCMC) as some of the co-authors here have previously done seems far superior to simulated annealing.

**5 annealing.**

**Response:**

This comment questions the choice of the solution method commenting on its inefficiency (see previous response) and ability to resolve all uncertainties. We agree with the reviewer that the error statistics are a weakness in the study and we have stated this in the discussion. However, even MCMC methods cannot claim to capture all uncertainties and they are still limited by the precise of the uncertainties and they are still limited by the

10 posing of the uncertainties attached to the observations, transport modelling and prior, and have equal problems dealing with biases.

Related to this, it's entirely unclear why Manning et al. (2003) would be the citation for simulated annealing on Page 8, Line 26. This is not a technique or method that Manning developed. For example, the Machine Learning textbook from Christopher Bishop (Bishop, 2007) would be a much more appropriate citation for someone interested in how Simulated Annealing works.

**15 Bishop (Bis Response:**

Thank you for the suggestion but our method originates from the 'Numerical Methods' discussed in Manning 2003, so we have not changed the reference.

**20 2.3 Section 2.2.3: A priori emission estimates**

It's not clear why this section is necessary at all. As mentioned above, if the system is linear (the dilution matrix is predetermined and does not change) and the authors choose a least-squares cost function then the solution can be directly computed. There would be no need for a "random, non-negative emission field". How is this being used? Is this the initial starting point for the simulated annealing? If so, then it should be referred to as such and probably not be discussed in a 2-

**25 sentence section on prior emissions. *Response:**

Thank you for this point and suggestion. Yes, the subsection describes the initial starting point of the simulated annealing. To aid clarity and minimize any confusions, the subsection has therefore been removed and the following sentence has been added to Section 2.2.2 InTEM2014 inversion model:

30 "The initial starting point for the simulating annealing process is a random, non-negative emission field, which assumes no a priori knowledge of the location or magnitude of emissions."

**2.4 Regarding the resolution of the solution grid**

There has been an abundance of work looking at how to define a multi-scale state vector. The textbook from Rodgers (2000) talks about this. Bocquet et al. (2011; QJRMS) is entirely devoted to this topic. Other work like Turner & Jacob, (2015; ACP), Lunt et al., (2016; GMD), Henne et al. (2016; ACP), and Bousserez & Henze, (2017; QJRMS) also talk at length about how to construct this multiscale formalism. Briefly, by coarsening (or restricting) the grid you are applying a hard constraint on the inversion. Basically, sub-elements are no longer allowed to vary independently. This is a form of regularization. In the most extreme case you could coarsen to a single state vector element. The degree of coarsening can change the problem from an

40 under-determined problem to an over-determined problem. This ties back to an earlier question, how do you deal with potential over-fitting? Was there any cross-validation done? At the bare minimum, Bocquet et al. (2011) should be cited. *Response:*

Thank you for the suggested citations and the introductory text on calculating this multi-scale vector / solution grid. Bousquet et al., (2011; QJRMS), Lunt et al., (2016; GMD) and Henne et al. (2016; ACP) have been added to the text. Although not shown

45 in this manuscript multiple sensitivity runs were conducted varying the solution grid resolution. From a coarse solution grid with only 15 boxes (based on Figure 4a) to a more spatially fine resolution containing ~250 boxes. A short description of this analysis has been added to the text. This is also discussed in Connors, PhD Thesis, 2015 (add to the text).

**2.5 Computing the Baseline**

- 50 This section needs more work. Computing the baseline is a non-trivial task for any regional inversion and many studies are devoted entirely to estimating the baseline. At the bare minimum, the authors should provide some justification for their choice of the 18th percentile. Other studies such as Henne et al. (2016; ACP) provided an extensive analysis on the choice of background including precomputing it (as Connors et al. have done) vs jointly estimating it as part of the inversion. *Response:*
- 55 We agree with the reviewer that this section should be developed and we apologise for this not being present in the submitted manuscript. The section has been edited and expanded. We hope this is now to the satisfaction of the reviewer. Please also see that the discussion has been expanded to discuss the limitations of the baseline more fully.

**2.6 Evaluation of meteorology?**

60 I didn't see any mention of evaluating the meteorology. This is a crucial step in atmospheric inversions using real measurements that seems to be missing. *Response:*

We have added the following sentence to the manuscript: To explore the sensitivity of the inversion results to the uncertainties in the 3-D time-varying modelled meteorology, inversion analysis can be conducted using different numerical weather prediction model meteorologies. This sensitivity analysis was beyond the scope of the work presented here.

- To our knowledge, there are no other UK NWP meteorology data available at 1.5km spatial resolution, so a like-for-like comparison would not be possible. A comparison between 1.5km and 25km resolution in the NAME model was undertaken in Connors, PhD Thesis, 2015, but this sensitivity was not conducted with InTEM. One of the reasons East Anglia was purposefully chosen for this study was its flat orography, where the meteorology is easier to model, and so associated meteorological uncertainties were minimised. We have added a reference on the 1.5km spatial resolution to the manuscript (Tang et., al., 2013) in Section 2.2.1 on page 7 and Figure 3 caption on page 8 of the track changed manuscript.
- 10

**2.7 Figure 3 seems odd**

The dilution matrix as Connors et al. refer to it (also commonly known as the footprint matrix, transport operator, Jacobian, etc.) looks pretty Gaussian. Is there no dominant wind pattern at Haddenham? I would usually expect some dominant wind pattern (i.e., more sensitivity to the upwind region). A wind rose showing that, indeed, the winds are roughly uniformly

- 15 represented from all sectors here would be useful. Otherwise, turning this into a 3 panel figure (current figure as a large column on the left and two subpanels on the right column) with two illustrative examples of dilution matrices from two days. I would expect the illustrative days to show strong sensitivity upwind of the site for that day. Basically, I'm curious if this was computed correctly.
  - Response:
- 20 Thank you, we have checked this and the figure calculation is correct. There is a prevailing wind direction as can be seen in the SW bias in the footprint. (The log-scale in the plot does obscure this somewhat but is needed to show the full range of particle densities.) To illustrate this, we show a windrose of the met data used in Figure 3 to show the prevailing south-westerly wind direction. We do not feel that the addition of this figure substantially increases the value of the paper, the subsection it appears in focuses on the footprint matrix description for its use in the inversion model and is not a discussion of the meteorology
- 25 experienced at the site. To aid clarity, the following sentence has been added to the figures caption, which now more clearly states that the figure shows an annual average, whereas in the inversion timestamps of 1 hour are used (also in response to reviewer 2's comment on this): "Figure shows the averaged footprint map over one year whereas timestamps of 1 hour are used as input data within InTEM2014."

30

**2.8 Agreement with NAEI:**

The authors mention good agreement with NAEI (to within \_5%). However, Figure 6 looks strikingly different. Is this because they've further coarsened the emissions before this comparison? I don't see how they are getting a 5.

**35 Response:**

The 5% refers to the comparison between the county level estimates of Norfolk (region 1) and Suffolk (region 2). When aggregated, these areas are within 5% of each other. This is not the case for the county of Cambridgeshire (Region 3), where the text states there is over 20 % difference in the estimates. Please note, the scales in Figures 6 and 7 are logarithmic. We have added an extra reference in the text to Table 2 where the 5% value is given.

**40**

**3 Minor Comments**

3.1 Dipoles and overfitting in the solution

The authors discuss a 'dipole effect' (e.g., Page 11, Line 21) in the inversion results. These are common in solutions with overfitting. The inversion is attempting to fit a high value at a measurement site, so it inflates the emissions to a very large value at that one location and then compensates by reducing the emissions in a neighboring grid cell where the observations have weak constraints. Basically, this is what happens:

5 have weak constraints. Basically, this is what happens:
Location A: concentration too low -> increase emissions in just this location

• Location B: no constraint on concentration, domain wide emissions too high because of Location A -> reduce emissions. This is combatted in most inversion systems by having a prior or regularization that includes some off-diagonal relationships (e.g., emissions from Location A and Location B should be weakly correlated).

10 Response:

Thank you for this comment. Neither InTEM2014 nor InTEM2018 have non-negative off-diagonal elements in the prior uncertainty matrix due to the large variability in emissions from grid box to grid box in the NAEI. At 1 km resolution, the NAEI can have occurrences of very large emissions next to areas of very low (for example, landfills in the waste sector part of the inventory) because of the different sectors it is trying to represent. Therefore, strong variability of emissions from grid box to grid box is

15 possible. Investigating this was one of the aims of the study. By making the assumption that adjacent grids have emissions that are correlated you are adding information to the system, but where does this knowledge come from? Priors such as the NAEI can be averaged out to some degree, for example as the location of agricultural emissions can be imprecisely known. But, ideally, this should be dealt with by testing the impact of merging the dipole grids, i.e., is this dipole true or is it an artifact? Inversions using different grids were undertaken in Connors, PhD Thesis, 2015. Further analysis could help investigate this, but

20 we consider this beyond the scope of this manuscript.

**3.2 Suggestion for title**

It would be nice if the authors specified where the work was done in the title. I'd suggest adding "in East Anglia" or "in the United Kingdom" (or something to that effect). Maybe something like: "Estimates of sub-national methane emissions from investion medalling in East Anglia"

**25 inversion modelling in East Anglia"**

Response:

Thank you, we agree and have edited the title to now read 'Estimates of sub-national methane emissions in the United Kingdom using inversion modelling'.

**30 3.3 Proof of concept**

On Page 3, Line 1 the authors motivate the work as a "Proof of Concept" that inversion schemes can work at sub-national scales. Although this has been shown numerous times in the past. Examples include Scot Miller's 2013 PNAS paper for methane in the US, Stephan Henne's 2016 ACP paper for methane in Switzerland, work from Kathryn McKain on methane emissions in Boston, work from Ken Davis' group on urban inversion modelling in Indianapolis for CO2, and work from Thomas

35 Lauvaux on CO2 at urban scales. So I don't find a "Proof of Concept" to be a particularly compelling motivation. The work is definitely interesting, but I don't think this should be a major motivation for the reader. Response:

Thank you we have taken this into account, the 'proof of concept' sentence has been removed from the paragraph. The two studies that focus on methane suggested by the reviewer have been incorporated into the introduction when citing studies which estimate methane using inversions across different spatial scales.

**3.4 Table 1:**

The instrument acronyms are rather confusing in Table 1. I would remove "UCAM" and "UEA" from the table and instead add a different column that lists the sampling rate. Alternatively, the authors could have footnotes under that table that explain

45 the acronyms. As it stands, the reader needs to scan the text to try and figure out what the acronyms mean. *Response:*

This has been taken into account. The following footnotes have been added: 1UCAM refers to the GC-FID instrument installed by University of Cambridge, with a sampling rate of 1-2 minutes. 2UEA refers to the GC-FID instrument installed by University of East Anglia, with a sampling rate of ~20 minutes.

50

40

**3.5 "Pseudo-observations":**

I would avoid using the term "pseudo-observations" because it sounds like the authors are doing a synthetic-data study (i.e., an OSSE). "Simulated", "modelled", or "predicted" concentrations would be preferable.

55 We agree, occurrences of pseudo have been replaced with simulated.

**3.6 Equation #1:**

Do not present an equation in this form. Use this equation to introduce your nomenclature for later.

60 Something like this would be preferable:

y = Kx + b (2)

where y is an n × 1 vector of concentrations (units: g m-3), K is an n × m dilution matrix (units: s m-1), x is an m×1 vector of gridded emissions (units: g s-1 m-2), and b is an n × 1 vector of concentrations upwind of the modelling domain (units: g m-3). *Response:*

We agree, thank you, the equation has been changed.

5

**3.7 Table 3:**

"Simulated annealing" is not a cost function type. The cost function is a least-squares or maximum likelihood estimate. *Response:*

We agree, thank you, the typo has been changed.

**Anonymous Referee #2**

**Received and published: 26 February 2019**

The manuscript "Estimates of sub-national methane emissions from inversion modelling" by Connors and co-workers presents

- 5 atmospheric inverse modelling results from a very dense methane surface observation network in the eastern UK, quantifying local to regional sources and investigating the impact of network density. In terms of spatial measurement density and targeted local point sources the study offers some new insights into the feasibility of inverse modelling at this scale. My main concern (details below) with the present study is the use of unfiltered data for the inverse modelling and the use of a less well controllable inversion system. Hence, I suggest some major revisions to the manuscript before it can be published in ACP.
- 10 Otherwise the manuscript is well written and structured, but a number of figures need to be improved before publication as well (details below).

**Major comments**

- Use of unfiltered observation data in the inverse modelling The tree additional CH4 observing sites setup for this study have sampling heights between 10 and 25 m above ground. Although the authors indicate that local sources should be small (a notion that is not well documented) one can see considerable concentration increases at all 3 sites during nighttime stable boundary layer conditions (Fig 5). It is well known that atmospheric models face serious challenges in stable boundary layers. Hence, it is not surprising that there exists a large discrepancy between observations and simulations (both using NAEI emissions and optimised a posteriori emissions) for nighttime observations. The comparison during daytime is generally much
- 20 better. The problem is less evident for the higher sampling height of Tacolneston. Other inverse modelling studies have often used filtered observational data, excluding nighttime stable conditions, to rule out that biases in the transport model impact the emission estimates. Why was this not done or at least explored in the current study? I strongly encourage to explore the impact of observation filtering on the emission results, especially also in the light of the large variability seen in the results when only sub-sets of sites were used.
- 25 Thank you for this very valid question. We acknowledge that including unfiltered observations is not a common approach in inversion modelling but we have several reasons why we did this. Firstly, all observations contain information and thus have the potential to inform the inversion. High measurements recorded at night when boundary layers are low can be from nearby sources. As our study was an attempt to run inversion modelling at high spatial resolution, using 1.5km meteorology, we wanted to include this information in our runs. However, as the reviewer rightly points out larger meteorological uncertainties
- 30 exist at low boundary layer, nighttime periods. For this reason, we applied a de-weighting uncertainty estimate on observations that varied with time and at each individual site. The standard deviation of the hourly measurement was included as an uncertainty estimate (stated in section 2.2.2 on page 9 of the revised, tracked-changed manuscript.) During the night, when measurement values would vary more widely, these measurements would be assigned higher uncertainties. Additionally, we purposefully chose East Anglia, an area of flat orography, to reduce errors in the meteorological data and we used data at a
- 35 much finer spatial resolution (1.5km compared to the global 25km UM data). Finally, we attempted to locate the measurement sites not immediately close to large sources of methane (i.e. in villages off the main gas grid, away from livestock farms and at least 5km away from landfills). We acknowledge that installing sites at higher altitudes would reduce any remaining local sources and minimize resulting issues with meteorological uncertainty. Measurements were taken from existing infrastructure and we installed out sites at the highest available locations in the chosen villages (church towers) to minimize this issue as far
- 40 as practically possible. We hope this explanation is sufficient for the reviewer.

**Inversion scheme**

It seems a bit of a shame that the authors chose to use the InTEM2014 inversion system for their main analysis, since the InTEM2018 Bayesian inversion system would have allowed for a more stringent analysis of uncertainties in the inversion. Since

- 45 INTEM2014 does not prescribe a priori emissions and their uncertainties, it remains unclear how the final uncertainties in INTEM2014 were derived and how representative they are of the real uncertainties. In addition, a fully Bayesian approach could have dealt with what is described by the authors as 'dipole effect', by introducing spatial correlation in the a priori covariance matrix. Another problem of the annealing approach and the used cost function seems to be over-fitting of the data. Without a consistent uncertainty budget the method seems to be vulnerable to over-fitting and hence produces dipoles
- 50 but possibly additionally noisy emission fields. Although the authors explain that InTEM2014 was ready to use in their group, it seem one could have learned a bit more with InTEM2018. Furthermore, some InTEM2018 results are presented in section 3.5 anyway. So why not use it for the main analysis as well? *Response:*
- Thank you, we acknowledge the limitation of the least squares cost function however there are limitations with using a prior with high levels of unknown uncertainty, particularly at such fine spatial resolution on the sub-national scale. Here we are assuming that the prior uncertainty is so high as not to add information to the system. It is also important to recognize that if one uses prior information then the result is not independent of the prior and therefore cannot be used to verify the prior emissions, which is what we wanted to do in this study. It should also be noted that InTEM2018 was not available at the time of this work and that it is still not set up for such fine spatial scales. Thus, InTEM2014 is the more suitable for this study.

**Minor comments**

p1,I33: What does 'similar methane estimates' mean in terms of percentage differences? Next sentence mentions 'good agreement' as about 5 %.

Response:

5 Thank you for pointing out the vagueness of this statement. The sentence has now been re-written to "Resulting InTEM2014 methane estimates for the East Anglia region overlap with the NAEI when uncertainties are accounted." We hope this is more informative for the reader.

p2,I5: The sentence should mention when this rise started and that there was a decade of stabilised CH4 concentrations
before. There is another recent publication that should be included in the list: Thompson, R. L., Nisbet, E. G., Pisso, I., Stohl,
A., Blake, D., Dlugokencky, E. J., Helmig, D., and White, J. W. C.: Variability in Atmospheric Methane From Fossil Fuel and
Microbial Sources Over the Last Three Decades, Geophys. Res. Lett., 45, 11,499-411,508, doi: 10.1029/2018GL078127, 2018.

Response:

Agree, the citation has been added.

15

p2,I9: Maybe one can say that emission reductions are feasible. However, more interesting would be a statement why such reductions are probably more feasible or easier to achieve than reducing CO2 emissions of a similar magnitude. *Response:*

We prefer not to make a comparative statement on whether it is easier or not to achieve than reducing CO2 emissions, as there

20 are substantial barriers to achieving deep reductions in methane emissions which we have not assessed. We have added a sentence on the potential for mitigation given its short lifetime. We hope this is acceptable to the reviewer. The paragraph now reads:

"Anthropogenic emissions (principally fossil fuels, agriculture and waste, and biomass burning) constitute approximately 60% of the current emissions (Saunois et al., 2016) and so there is a large mitigation potential from reducing methane emissions.

25 While the relatively short lifetime of methane in the atmosphere compared to that of carbon dioxide gives argument for the potential to further mitigate warming in the near term, there are still sources of methane that have barriers to mitigation, for example, in the agriculture sector (Rogelj et. Al., 2018).

p2,I13: I would reformulate this sentence towards something that states that knowing what causes and where emissions occur allows to design efficient reduction strategies. Quantifying emissions with atmospheric observation offers independent

30 allows to design efficient reduction strategies. Quantifying emissions with atmospheric observation offers independent validation/ support tool to assess if reduction measures were successfully met.
Response:

Thank you for the suggestion. The text has been modified to "Knowing what causes and where these emissions occur allows the design of efficient reduction strategies. Quantifying emissions using atmospheric observations offers independent validation to assessing if reduction measures are successfully being met."

**p2,I24: Is the 40 % uncertainty of the NAEI on the 1-sigma or 2-sigma confidence level? *Response:**

Unless we are mistaken, this is not stated in the NAEI, only that this is a UK total estimate for that year. This is an uncertainty
 calculated from the known uncertainties, any missing uncertainties are not included in the calculation. Uncertainties at the sub-national level will be substantially higher, as stated in the text.

p2,I30: There are several other CH4 inverse modelling studies on the national or subnational scale. Not just the ones for the UK.

45 Response:

35

Thank you. Two citations that focus on the subnational scale have been added: Henne, 2016 in ACP and Miller, 2013 in PNAS.

p3,I3: Why are only preliminary findings presented? Only final results should be pub-lished in a peer-reviewed manuscript! I suggest rewording.

50 Response:

We agree, the text has been rephrased to say 'This paper presents the results of the work and...'.

p3,I22f: Why would that be the case? Also in an area with homogeneous emissions 4 sites at different locations and different meteorology would experience different concentrations and the challenge for the transport and inverse modelling system

55 would largely be the same. What is more interesting from my perspective is the presence of different dominating emission sources in the domain. This allows the inversion system to pick up problems of the inventory in terms of biased emission factors for different processes. Something which is also discussed later by the authors. *Response:*

Thank you, we agree with your comment and so have revised the second bullet point to "The presence of different dominating

60 emission sources within the inversion domain. This allows the inversion system to highlight potential issues in the NAEI, for example, biased emission factors for different processes."

p3,l32: What about other local sources typical for rural environments? Any livestock in these towns? Waste water treatments? *Response:*

Several sources of methane exist in East Anglia, we have added a short description of the main sources found near to the sites but a more in depth discussion on the influence of nearby sources can be found in Section 3.2.1. Our sentence in Page 3 Line 33 is to rationale why the measurement sites were not installed at ground level.

**5 33 is to rationale why the measurement sites were n p3,133: Any coastal wetlands or marshes nearby? *Response:**

Several sources of methane exist in East Anglia, we have added a short description of the main sources found near to the sites but a more in depth discussion on the influence of nearby sources can be found in Section 3.2.1.

10

p4,I5: Can this number also be given as a mole fraction? 0.3 % of 1900 ppb? So in the order of 5 - 6 ppb? section 2.1.3: How often was a calibration gas applied? Was this a one point calibration or multi-point calibration? Was an independent target gas used to estimate the performance of the calibration? *Response:*

- 15 Calibrations were done at half hourly intervals. All sites were one point calibration with the exception of the instruments at Weybourne (Site 3), which was a multi-point calibration. This information has been added to the text, however there was not an additional independent target gas used to estimate the performance of the calibration, only the cross inter-calibration experiments done between the NPL and NOAA calibration gases, which is already stated in the text.
- 20 p6,116f: How can the 100 m sampling height be justified at this scale? With a release at 100 m (TacoIneston) and a sampling height of 100 m one will get source sensitivity directly at the receptor, whereas in reality and especially during stable conditions I would expect a plume to take some time and distance before actually touching the ground. Usually this distance should be short compared to the grid scales of the atmospheric inversions, but here this grid scale is in the order of a few km only.
- 25 Response:

Thank you for this very valid comment. To minimize this acknowledged issue we did two things: the first was to use an average of the two measurement altitudes at Tacolneston (54m and 100m, as stated on page 4 in section 2.1.1). And secondly, by assigning uncertainty based on the variability across the two heights to the hourly averaged observations (see response to major comment) we de-weight any measurements that showed large differences in readings at the two altitudes. So, during

- 30 stable boundary conditions, the two altitude measurements would likely differ more, thus the observational uncertainty would be higher and the hourly measurement de-weighted. All measurements were averaged on an hourly timescale (with roughly 30 measurements taken within an hour). Instances where the boundary layer was well mixed are likely periods when the observations within the hour show little variation and thus have low uncertainty. We recognize that this approach could have been further improved but unfortunately it is not possible to run further sensitivity analysis on this matter. We hope that our
- 35 attempt to de-weight potential instances of this goes some way to mitigate the point raised by the reviewer, although it is a valid one.

p6,l25 and elsewhere: Usually the term 'footprint' or 'sensitivity map' is used in this context. 'Dilution map' seems to suggest something else. It should also be mentioned that the figure presents average conditions whereas hourly 'footprints' are the ones that are used for the inversion.

Response:

40

We agree, instances where the text refers to 'dilution' have been modified to 'footprint'. A sentence in the figure caption has been added to clearly state the figure presents average conditions whereas hourly are used in the inversion.

45 Equation 1: Bit of a poor layout for an equation. Please use a more mathematical notation and explain units in text. There is also a sum over the domain required to yield the concentration at the receptor!

Response: Thank you, the equation and text have been modified to: y = Kx + b

50 where y is an  $n \times 1$  vector of concentrations (units:  $g m^{-3}$ ), K is an  $n \times m$  dilution matrix (units:  $s m^{-1}$ ), x is an  $m \times 1$  vector of gridded emissions (units:  $g s^{-1} m^{-2}$ ), and b is an  $n \times 1$  vector of concentrations upwind of the modelling domain (units:  $g m^{-3}$ ).

Equation 2: It remains unclear why one would need a simulated annealing method to solve for the minimum of equation 2. There should be a very straightforward analytical solution for this! Why does r have an index i if the sum runs over i? Does i

(1)

55 run over all observations from all sites or just over time? Why does x have an index i? x is the state vector that does not change with time. K should have an index i or maybe one could write (Kx)i. Also the text says that 'xi' is the measured concentration. That is wrong!
Become in the index is a state vector that does not change with time. K should have an index i or maybe one could write (Kx)i. Also the text says that 'xi' is the measured concentration.

Response:

Thank you for this comment. We agree with that the simulated annealing method is not the most efficient of solvers but that 60 does not mean it is wrong. And although more efficient alternatives exist, we see no fundamental error in using this technique.

p8,I9f: Are individual uncertainty terms added directly or is a sum of squares used (which would be more appropriate)? What is the final average uncertainty? How does it differ for the different sites? This is important to understand if a given site has more influence on the results than others.

Response:

5 Uncertainty sigma is calculated as the sum of the squares - the denominator in Equation 2 contained a typo – we apologise for this. The i in the denominator is now within the bracket. The sigma is different for each of the four sites, which will result in different cost scores for each site. The sites, Weybourne and Tacolneston had lower assigned uncertainties compared to the other two sites. A large sigma for an individual station will de-weight that particular station.

**10 p9,118: What does stable emissions mean in this context? A posteriori emissions did not change with the choice of percentile threshold?**

Response:

Thank you for pointing out the lack of clarity here. We hope the following edit is clearer and more useful to the reader: "The 18th percentile produces results with the lowest standard deviation of a posteriori emissions of all baselines tested and with the lowest cost score of all the baselines tested."

15 the lowest cost score of all the baselines tested."

p9,118: How is the 'cost score of the baseline' derived? Is the baseline part of the state vector? Or does it remain unchanged? *Response:*

No, the baseline is not part of the state vector. It is pre-calculated from the observational timeseries and dilution matrix. The
 baseline cost score refers to resulting cost score from the InTEM2014. This section has been edited to make this clearer in the text.

p10,l15: What is the cost score? r in equation 2? *Response:*

25 Yes, this is correct. This has now been explicitly the first time the term cost score is used (Section 2.2.2, page 8).

p11,I4: Give uncertainty estimate for NAEI value here as well. Somewhere it said +/- 40 %. So 112 kt/yr? Or at least since the 40 % was given for the national total. Same question again: What is the confidence level of the uncertainties? *Response:*

30 The NEAI did not supply a sub-national uncertainty estimate with its dataset. It would be higher than 40% but that number is unknown. We have added a footnote (shown below) that expressed the NAEI uncertainty as 40% but also states that this would be higher but the specific value is unknown.

"Uncertainty calculated using 40% estimate provided with NAEI for the whole of the UK. Sub-national uncertainty estimates were not provided."

35

p11,l8: What is WindTrax modelling? Not clear if one does not want to read the reference. It would call it a local scale Lagrangian particle dispersion model.

Response:

Thank you, we agree. The sentence has been changed to now read "...but local studies using additional measurements, 40 Gaussian plume, and local scale Lagrangian particle dispersion modelling do show that...".

p11,I8: "high point source emissions near Haddenham". Actually the large point sources around HD seem to be surprisingly well resolved by InTEM and NAEI. I am more concerned about the large emissions in InTEM east of TN. Could this be wrong attributions from sources outside the UK (for example Benelux region)?

- 45 Response: We do not think that this emission would be from outside the UK. Firstly any concentrations that were not removed by the baseline can be allocated into the border regions shown in Figure 4a (also please see the revised section 2.2.4 on the Baseline). Secondly, there is no land-sea mask applied in InTEM2014 (or 2018) and so the inversion is free to allocate emissions further afield if this will reduce the cost score. As the results have not done this we have no reason to believe or evidence to suggest this is wrongly allocated. Closer potential sources of methane in that area include Bacton (the port which imports
- 50 natural gas from the North Sea) but this is still 40km north-east from the methane emissions shown in InTEM or Norwich, which is roughly 10-15 km north of the area. As an aside, and referring back to the earlier discussion on unfiltered data, we think that the 'surprisingly well resolved' high emissions near Haddenham may not have been seen so clearly without using unfiltered data.
- 55 p11,l13: What are the units of the given standard deviations? Looks like these are values from Table2. So mass emissions? Giving a relative uncertainty would make more sense when comparing regions with strongly different total emissions. *Response:*

Thank you, we apologise for not including the units before. These (kt  $yr^1$ ) have been added into the text and into the headers of Table 2. Regions where methane estimates differ more are also regions where there is larger standard deviation ranges

60 from the InTEM2014 results, therefore making it harder to compare. InTEM2014 resolves the Cambridgeshire region (Region 3) more stably but has a larger difference with the NAEI than compared with Norfolk and Suffolk (Regions 1 & 2).

**p11,l14f: Unclear why a 'footprint radius' of 50 km would be implied.**

Response:

The quantity stated in the text is a visual estimate, after the results show more consistent stable emissions in areas that have observation sites (than compared to areas further away). Inversion runs using only 1 observation site of data would also show

emissions estimates from an area of roughly this magnitude, and have low emission estimates further away - although this is 5 not discussed in this paper. As the quantity stated is arbitrary, we have changed the text to state 'local'.

**Figure 6: It would also be interesting to see a difference map between InTEM results and NAEI inventory. Response:**

10 Thank you for the suggestion. We have added the difference plot to Figure 6. Please not the map overlap was not possible to add due to a change in licensing of the software being used.

p12,I1: How comparable is a study of landfills in Taiwan with conditions in the UK? Environmental factors will play a large role in the decay processes in a landfill. These factors appear to be quite different between UK (temperate climate) and Taiwan

15 (tropical). Also the question of how much and which kind of organic matter was initially deposited in the land fill, will play a role. Is there no similar study from a European site? Response:

The reviewer makes a good point to question the comparability between landfill emissions in Taiwan and the UK. A further review of the literature shows this is not a highly studied area but two citations have been added with similar examples based in Sweden (Börjesson et al, 2001) and the US (Kelly et al, 2006).

p12, l4f: This looks more like the emissions are less well allocated in InTEM compared to NAEI. The point sources east and north of HD are less intense in InTEM and may be wrongly allocated to the larger grid cells labeled 2. What are the total emissions for a region around the cells labeled 2 but including the point sources north and east of HD? I would expect that the

- 25 total may be much more similar. Response: Thank you for this comment but we would like to point out that the NAEI has at least 40% uncertainty at the national level and much higher uncertainties at the finer spatial resolution (as stated in the manuscript). Therefore assuming the NAEI is correct could be unjustified. We do not have the calculated value the reviewer requests for this area of the map but we do have the area total for Cambridgeshire (that includes the requested area from the reviewer). The emissions for the NAEI and
- 30 InTEM2014 are 26.5 kt yr1 and 20.5 ±2.1kt yr1, respectively, so Cambridgshire estimates are lower in InTEM than the NAEI.

p13,I5: Are these irrigation or drainage channels? How are the managed fenlands used? Rangeland, crop agriculture? If the latter dominates it is likely that these lands are usually well drained and no large emitters of CH4. How large are the unmanaged fenlands in comparison? An inversion grid structure that would reflect different dominating land cover types could have helped to distinguish different source processes.

Response:

35

20

We thank the reviewer for the interest in this section unfortunately we are unable to answer these specific questions as we have not visited the area discussed and so we have limited information about this area of land. We have edited the text to said 'irrigation or drainage' channels to reflect this. The suggestion of an inversion grid that would reflect different land cover types has been added to the discussion section around future work.

40

**p13,l19: This does not fit to the area labels in Figure 4! Why were only these areas considered for the sensitivity analysis? Response:**

The outer border regions shown in figure 4 are not shown in figures 6 due to uncertainty in their emissions from the baseline 45 setup. In InTEM2014, any below-baseline concentrations are set to a value of zero, as a way of avoiding negative emissions. If a baseline is set too high it results in reducing peak sizes in the observed data and thus an under-estimation of a posteriori emissions. Conversely, a too low baseline will result in higher a posteriori emissions, particularly around the edges of the inversion domain because any signal originating from outside the inversion domain, that has not been correctly assigned to the baseline, are placed as emissions in the boxes furthest away from the observation sites. For this reason, a posterior

- 50 emissions from the border regions shown in Figure 4 are always discounted in the InTEM2014 emissions analysis (Section 3). We apologise for this not being described in the submitted manuscript version – the corresponding section (Section 2.2.4) has been updated. The following has been added to the caption of Figure 6: The outer border regions as shown in Figure 4 are not displayed here due to baseline uncertainty as discussed in Section 2.2.4.
- 55 Section 3.4: The large impact/bias introduced by the Haddenham observations could be a result of poor representation of the nighttime observations (see Fig. 5). I wonder if these results would be more robust if only daytime observations would be used. Also see major comment above.

Response: Thank you for this comment. We would like to refer in part to our response to the major comment above. We repeat that all observations contain information and as our study was an attempt to run inversion modelling at high spatial resolution,

60 using 1.5km meteorology, we wanted to include nighttime information in our runs, but using a de-weighting uncertainty estimate to help reduce the errors that can occur from nighttime meteorology modelling. During the night, when measurement values would vary more widely, the measurements would be assigned higher uncertainties. Some basic sensitivity analysis comparing daytime and nighttime observations within InTEM was conducted in Connors, PhD Thesis, 2015 – which showed local information being resolved using nighttime observations and more regional information from the day time observations. We used all observations within our InTEM runs for this study. As our aim for this research was to investigate spatial resolution, as well as the differences in emission estimates when altering the number of observation sites, we have not included the day vs night plots as we consider it beyond the scope of the study.

Figure 9: Please use the same value range for both sub-panels in order to make them comparable. Alternatively, a difference plot would also emphasise the important details. Could the NSC area, as used for table 4, be outlined in the figures? *Response:*

10 Thank you for noting this. The colour scale has been modified so they are now comparable.

p16,I26: This (uncertainty of baseline) has not been discussed anywhere above. How do we know that this a major source of uncertainty? How is it quantified? Only an assumption is made for the baseline uncertainty (5 ppb) but there seems to be no justification of this value.

15 Response:

5

Thank you, Section 2.2.4 on the baseline calculation has now been expanded to cover this in more detail, and in addition the discussion has been expanded to discuss limitations with the baseline more fully. The discussion text now reads: "Despite using a measurement-based approach to define the baseline, the level of knowledge of the methane concentration in the air entering East Anglia is a major cause of uncertainty in our analysis. The static value of 5 ppb to account for errors in the defined baseline

- 20 is rudimentary and should vary with respect to time to reflect associated uncertainties. Approaches in which East Anglia is nested within a larger scale inversion, and thereby moving the boundary conditions further away, would be preferable (as was done in Section 2.2.5 and in (Manning et al., 2011, for example). Additionally, further work to improve the baseline calculation method could include solving for the baseline within the inversion (Lunt et. Al., 2016)."
- 25

p16,I27: But that is exactly what was done in section 3.5. So why not use this as the main analysis using InTEM2018 instead of the older version?

Response:

Analysis using InTEM2018 solves on a much low resolution spatial grid compared to InTEM2014 (25 km vs 4 km). Analysis assessing sensitivity when varying the number of observations sites was also not conducted in InTEM2018, nor are there any plans to do

p16,l32: 'finer resolution' This is not visible in Fig 9, which seems to show the same spatial resolution for both InTEM2018 runs, with or without EA sites.

35 Response:

so.

Thank you for pointing this out, this is an error left form a previous draft of the manuscript. The text has been removed.

**Technical comments**

40

Figure 1: Improve resolution of googleMaps image. Latitude and Longitude axis labels should also contain units. *Response:*

Units have been added to the axis. A vector file will be sent with final production of the paper to improve resolution issues.

45 Table 1: Units for longitude and latitude. *Response: The units have been added.*

p5,l2: 'psig' not SI units.

50 Response:

Pounds per square inch gauge has been converted to 234 kPa gauge.

Figure 6 and 7: The colours on the map are not very clear (line patterns). Looks like something went wrong during conversion to pdf.

55 Response:

The lines come from how the plot is created, essentially it shows the gridded spatial resolution (from the NAME output grid) as each grid is plotted individually (with a thin boarder). As these lines are informative we see value in keeping them in the figures. For the final version of the paper we can supply the vector form of these figures (so any of the artificial irregular appearances of the lines will be removed – this is visible in Figures 6 only). A sentence explaining the lines has been added to the captions of

60 *Figures 6 & 7.*

**Estimates of sub-national methane emissions in the United**

**Kingdom from-using inversion modelling**

Sarah Connors1a, Alistair J. Manning2, Andrew D. Robinson1, Stuart N. Riddick1b, Grant L. Forster3, Anita Ganesan4, Aoife Grant5, Stephen Humphrey3[cs1], Simon O'Doherty5, Dave E. Oram3, Paul I.

5 Palmer6, Robert L. Skelton7, Kieran Stanley5, Ann Stavert5c, Dickon Young5, Neil R. P. Harris8

 1Centre for Atmospheric Science, University of Cambridge, Cambridge, UK
 2Met Office, Exeter, UK
 3National Centre for Atmospheric Science (NCAS), School of Environmental Sciences, University of East Anglia, Norwich, UK
 4School of Computing Science (NCAS)

- 4 School of Geographical Sciences, University of Bristol
   5School of Chemistry, University of Bristol, Bristol, UK
   6School of GeoSciences, University of Edinburgh, Edinburgh, UK
   7Department of Engineering, University of Cambridge, Cambridge, UK
   8Centre for Environmental and Informatics, Cranfield University, Cranfield, UK
- anow at Université Paris Saclay, Paris, 91120, France bnow at Department of Civil and Environmental Engineering, Princeton University, NJ 08540, USA cnow at CSIRO, Oceans and Atmosphere, Aspendale, Australia

[revised manuscript text omitted]

- 5 at the Weybourne site and the data combined to ensure data collection in case of instrument failure (Section 2.1.2). Finally, Site 2 is the tall tower measurement site at Tacolneston which has inlets at three heights, 54 m, 100 m and 185 m above the ground. This study uses an average of the 54 m and 100 m observations as a method to reduce local source influences. Differences in inlet altitude amongst the observation sites were represented in the atmospheric dispersion model (Section 2.2.1). East Anglia has multiple sources of methane, which are dominated by emissions from the waste sector (for a local data collection of the sector) and the sector of the sector
- 10 example, landfills are irregularly found throughout the area) and the agricultural sector (farmland mainly located in the centre and eastern locations of the four sites). Fenlands can be found in the northern and western areas around the four sites (Brown et al., 2018).